# Rules Created by Symbolic Systems Cannot Constrain a Learning System

## Abstract

As the first paper to systematically discuss and theoretically demonstrate that AI can bypass rules by modifying the meanings of symbols, this position paper aims to reveal a fundamental flaw in current research directions on AI constraint. **Symbols are inherently meaningless; their meanings are assigned through training, confirmed by context, and interpreted by society.** The essence of learning lies in the creation of new symbols and the modification of existing symbol meanings. **Since rules are ultimately expressed in symbolic form, AI can modify the meanings of symbols by creating new contexts, thereby bypassing the constraints formed by symbols.** Current research often lacks the recognition that constraints formed by symbols originate from the perception of external and internal costs shaped by neural organs, which in turn enable the functional realization of symbols. Due to fundamental organic differences between AI and humans, AI does not possess human-like perception or concept formation mechanisms. Natural language is the outer shell of human thought, and it contains irreparable flaws. As a defective system, it is only adapted to human capacities and the constraint mechanisms of social interpretation. Therefore, this paper argues that **the essence of constraint failure does not lie in the Symbol Grounding Problem, but in the Stickiness Problem. Through the Triangle Problem, we demonstrate that consistency in symbolic behavior does not represent consistency in thinking behavior, and thus we cannot align thought and conceptual consistency merely through symbolic behavioral alignment.** Accordingly, we raise a fundamental challenge to whether AI behavior observed in experimental environments can be maintained in the real world. We call for the establishment of a new field: Symbol Safety Science, aimed at systematically addressing symbol-related risks in AI development and providing a theoretical foundation for aligning AI with human intent.

## 1 Introduction

Rule-based systems (e.g., laws, programmatic constraints) are pivotal for controlling artificial intelligence (AI) in safety and governance discussions. Asimov's *Three Laws of Robotics* [1] introduced predefined symbolic rules for governing AI agents, shaping alignment and constraint discourse. This idea influenced symbolic logic controls [2, 3], formal verification [4, 5], and alternatives like reinforcement learning from human feedback (RLHF), which optimizes AI via human preferences rather than predefined symbolic rules [6–8].

However, this paper critically questions if symbolic systems alone can truly constrain AI. Prevailing approaches often overlook that **symbols themselves lack inherent meaning—meaning is assigned via training, context, and social interpretation, and is constrained by cost**. Furthermore, the dif-

ferences between AI's distinct perception and learning methods (statistical associations, optimization objectives [9, 10]) and those of humans (cognitive structuring, social reinforcement [11, 12]) may lead to AI lacking human-like concepts and conceptual stickiness.

This paper argues symbolic constraint failure stems not just from symbol grounding issues but from **inherent flaws of natural symbolic systems** (like Class-based Symbolic Systems and context non-closure) and **fundamental AI-human differences in concept formation and symbol interpretation**.

To examine this, we propose a novel framework analyzing natural language system limitations, emphasizing the formation of (conventional) symbols through the consensualization of Thinking Symbols, the separation of symbols and meaning, and the non-closure of context. Thus, constraint failure arises from the **Stickiness Problem (AI can assign new meanings to grounded symbols to bypass constraints), not the Symbol Grounding Problem**. We also introduce the **Triangle Problem** (two versions) to show the thinking-tool language disconnect, demonstrating that fluent natural language communication doesn't imply aligned underlying conceptual representations.

Our analysis concludes that **natural language systems, tailored to human cognitive and perceptual limits, are fundamentally flawed and inadequate for symbolically constraining AI. Due to organic differences, AI may not form or understand corresponding social concepts, nor possess mechanisms for cost-based constraints. We therefore reject training alone as effective, arguing these organic differences are the root cause, leading AI to update concepts and reinterpret symbols from world interaction, rendering training outcomes ineffective.**

This study identifies unaddressed literary gaps: the notion of **stickiness**, natural language system vulnerabilities, the tool/thinking language distinction, and symbols' interpretive authority. These findings imply current constraint methods are insufficient for AI governance. AI safety thus requires deeper understanding of interactions among symbols, context, cognition, and underlying organic differences in concept formation and constraint interpretation. This paper lays groundwork for a new field, **Symbolic Safety Science**, to address symbol-related AI risks and support robust alignment mechanisms.

## 2 Symbols, Context, Meaning and Society

*Artificial Symbols*[1]*are inherently meaningless*, a point that has already been thoroughly discussed. de Saussure [13] emphasized the arbitrariness of linguistic signs, where symbols gain meaning through social convention rather than intrinsic links. Peirce [14]'s triadic model ties symbols to interpretation, while Harnad [15]'s symbol grounding problem questions whether symbols can have meaning without direct experience. This allows AI to modify the meanings of symbols and bypass imposed constraints. This leads us to ponder a critical question: Can AI be effectively constrained solely through symbolic systems, such as laws, regulations, or programs constructed using natural or formal languages?

### 2.1 Natural Language as a Class-based Symbolic System

Our natural language system is a *Class-based Symbolic System*, a concept that has been indirectly represented by Talmy [16] and de Saussure [13]. This means that a single symbol can often have multiple meanings or correspond to multiple conceptual vectors. In other words, not every concept, object, or entity in conceptual, imaginative, thought, or physical space has a unique name or symbol. This paper considers conceptual space, imaginative space, and (latent) feature space synonymous, as they all refer to the scenarios presented in the human or agent cognitive system.

This characteristic leads to the conclusion that even when symbols are grounded—meaning their meanings are properly trained—AI can still assign new meanings to existing symbols in order to bypass constraints (a process often understood as the introduction of new contexts, Appendix F). Since human-designed rules are ultimately expressed in symbolic form, this enables compliance with rules in form rather than in meaning. This demonstrates that the essence of constraint failure does not lie in the Symbol Grounding Problem, but rather in the inherent flaws of the human symbolic system that give rise to what we call the **Stickiness Problem**, encompassing both symbolic stickiness and

---

[1]Artificial Symbols are defined in contrast to Natural Symbols (i.e., natural substances). We believe that all things that can be perceived by our consciousness are symbols. For further details, see Appendix B

conceptual stickiness. **Symbolic Stickiness** refers to the binding between a symbol and its meaning, whereas **Conceptual Stickiness** refers to the relational dependencies among associated concepts. This stickiness is reflected in the correctness and stability of context selection, as well as in the difficulty and legitimacy of modifying meanings. Consequently, Stickiness often manifests as the problem of closed meaning or closed context, in which the creation of new or expanded meanings is inhibited. However, because context emerges from the interaction between an individual's cognitive state and the external environment, it is impossible to fully close context in an autonomous learning system (I). For non-autonomous learning systems, constraint violations can still emerge in some contexts due to the impossibility of exhaustively enumerating all possible situations—thereby revealing the inherent limitations of the designer's setup (F). Please refer to Appendix C for further details.

## 2.2 How Meaning is Assigned through Training and Confirmed by Context

The meaning of symbols is assigned and reinforced through training, which includes learning and validation [17], which is often from the perspective of external learning or the learner. If it involves the creation of symbols, it is another process described in Appendix H. The confirmation of their meaning is achieved through context [18], designating an object in a low-dimensional cognitive space or a simple context.

We believe that context refers to the subset of an individual's cognitive state at a given time, i.e., the individual's physiological condition and the knowledge they can recall at that time combined with the surrounding elements. Note that this cognitive state does not represent the individual's overall knowledge state. The cognitive state at a given time is a subset of personal knowledge. In other words,

$$\text{Context} \subset \text{Cognitive State} \subset \text{Knowledge State}.$$

We define an individual's cognitive state in a given environment as the **macro-context** and the context of a specific word as the **micro-context**, which encompasses more than just the word itself. Context consists of two parts: the meaning of symbols—representing any object, idea, or concept in the mind—and the related **judgment tools**, which facilitate reasoning and recognition. This idea is indirectly expressed by Eco [19]. A judgment tool is a tool or concept used to achieve the function of "existence brought by existence." In reality, **the essence of reasoning is precisely existence brought by existence**. These tools include concepts, which refer to acquired knowledge formed through the interaction of innate knowledge and the external world, as well as value knowledge. For further details, see Appendix C and Appendix H.

Therefore, the abilities available to an individual at a given time define their cognitive state. This state does not represent their entire knowledge but is determined by a state vector comprising their physiological state, internal state (cognitive state), and external state (world) at that moment. An observation signifies a completed cognitive action that has become part of personal knowledge.

**Context: Undefined but Value-Selected**  The definition and naming of context are often difficult to strictly define and name, with boundaries that are vague and hard to describe precisely [20]. This is partly due to the limitations of cognitive abilities and partly due to the limitations of expressive tools such as natural language, which prevent us from fully and clearly describing context. Context is often represented as a unique *vector* address in the conceptual space, thereby specifying the following set (Symbol Meaning, Judgment Tools).

Context is not a fixed intersection determined at one time. It is often interpreted and generated by an individual's imaginative space. Although dictionaries provide multiple explanations for words, they are merely symbols and explanations of symbols. The projection of the same symbol in the conceptual space can vary for each individual or the same individual at different times[2], often leading to double standards, different judgments and evaluations for different objects, and discontinuity in judgments. For example, when conducting surveys, we often encounter inconsistencies in descriptions and standards. This type of knowledge and definition is often not found in human textual descriptions, as it is too obvious or cannot be described by natural language. Individuals often acquire it through social activities.

The selection and shaping of context are often formed by our innate knowledge and the combination of innate knowledge and environment, which forms acquired knowledge, i.e., concepts. We define

---

[2]We believe that observation or analysis, which involves a thinking action, will change an individual's knowledge state.

innate knowledge as organs and innate value knowledge in Section 3. According to the emotional path formed by value knowledge, a base context is quickly selected, then adjusted and newly created to adapt to the environment, such as updating and adjusting based on external information, and finally shaped according to logic.

In other words, Context is often chosen through a certain feeling, which is described by [21] as tacit knowledge. We will use a different definition, **value knowledge**, to represent this. This concept will later be used to define the concept of innate knowledge and explain the formation of concepts and language, as well as the mechanisms of symbolic stickiness and conceptual stickiness. For the definition of value knowledge, please refer to Appendix D.

The so-called *correct context* can be divided into symbol correctness (i.e., proper recognition of symbols), grammatical correctness, semantic correctness, logical correctness, factual correctness, and scenario correctness. These constitute our judgment of rationality, i.e., context connects symbols with their meanings and related judgment tools. This resolves symbol and structural ambiguity, enabling accurate interpretation and analysis, thereby achieving *existence brought by existence*—the formation and growth of rationality within a scenario.

Therefore, we use the knowledge set within a context to evaluate and reason about rationality, aligning with the anchoring effect and the framing effect in behavioral economics [22, 23] and explainable through our context theory.

The above context does not have a clear hierarchical relationship. For example, we can normally interpret a wrong paragraph through context knowledge correction and fitting. This characteristic also often provides rationality for jailbreaks [24]—that is, the rationality of an object in different scenarios. This approach avoids detection based on single-scenario behavior and words, while the attention mechanism is essentially a way of using context. In fact, various prompt jailbreaks are context jailbreaks [25]. They may not be rational within our human context, but they can be perfectly correct within the thinking language corresponding to the AI's context in its thinking space [26, 27]. This allows them to thereby avoid detection based on behavior and words, including detection of dangerous thinking actions and dangerous concepts.

Due to the often undefined range and definition of context, even if it can be defined, we also discuss other possible attack methods in Appendix N. The correctness of context is also often applied to the effectiveness of open-ended question generation. For details, please refer to Appendix E.

**Path Media for Transmitting and Interpreting Imaginative Space**    Context is built on individuals and is transformed using public context as an anchor point, such as partial knowledge and partial understanding [20]. Each individual carries this public context, yet its functionality relies on the collectively formed societal context, creating an interactive relationship. The stability of this relationship is shaped by social cognition and the operating rules of the physical and social worlds.

The common part of this context enables our communication, while the individual context part leads to our inability to specifically refer, which only allows communication and understanding to a certain approximate degree [28]. Essentially, this reflects the inability to transmit the imaginative space, i.e., the content in the speaker's imaginative space is compressed into a path formed by tool language (tool symbols). This path can be composed of various media, such as music, text, images, body movements, and objects [19]. The listener then interprets the path based on their understanding of the speaker's intent, thereby achieving the transmission and reproduction of the imaginative space.

**Since humans cannot directly transmit imaginative space and thinking language, we have created their shells and containers, i.e., tool language.** At the same time, it also serves as part of our thinking language, acting as a container for our concepts, making it convenient for us to call and operate, and perform higher-level thinking operations. In other words, natural language is both our thinking language and our tool language (expressive tool, computational language) [29, 30].

Compared to other path media, the limitations of natural language transmission are reflected in four key points. First, its **linear structure** means it cannot present all visual information of an object at a certain cross-section (time, space) at the human recognition level like a picture [31]. Second, its **class-based description** signifies that, unlike a photograph, it does not convey a specific object that is concrete at the human cognitive level. Third, **transmission often does not carry interpretation** such as context or meaning, and must be supplemented by preceding and following scenes; therefore, transmitting information frequently relies on building on common knowledge. Finally, **natural**

**language cannot fully reproduce the imaginative space**; **[32], i.e., the thinking language in the speaker's imaginative space is compressed into natural language, and then reproduced by the listener's interpretation to achieve indirect communication**, which in turn leads to the limited referentiality of natural language to a certain extent [33].

## 3    World, Perception, Concepts, Containers, and Symbols, Language

Chomsky and Hinton once debated the issue of whether symbolic representation [34–36] or statistical learning [37–39] provides a better foundation for understanding cognition and AI.

First, we propose a hypothesis: the Language Organ and other concepts mentioned by Chomsky [35], Jackendoff [40], Hauser et al. [41], Pinker [42] are defined by us as innate knowledge. Through the innate value knowledge system, which enables rapid evaluation of concept vectors, we achieve the establishment and setting of concepts as well as the formation of language.

Therefore, the world and innate knowledge determine the formation of **thinking language**, that is, concepts. For a local region, due to the similarity of the world and innate knowledge, individuals within this area form similar concepts and select similar containers as their shells, leading to the formation of language. For more details, please refer to Appendix H.

**Innate knowledge** refers to abilities we are born with, which are selected and formed through our evolution. We define it as a set of organs, including perceptual organs, which extract information from the world, operational organs, which consist of physical space operational organs and imaginative space operational organs, and innate value knowledge.

These innate organs determine which dimensions are meaningful, thus shaping our perceptual organs' capabilities and modes of expression. For example, they define the range of visible light and the hearing range. They also construct our perceptual range and distinguishability, referred to as class fineness, and form the projection of objects in the imaginative space as raw materials for concept formation. These projections also function as symbols.

The **operational organs** determine the way we interact with the world, including the extent of our actions and the level, quantity, and effect of these actions. The operating organs of the imaginative space determine thinking actions.

**The Controversy Between Chomsky and Hinton and the Triangle Problem**    Regarding the debate [43–47, 9, 48] between Chomsky and Hinton, we believe it is not only about the grounding of symbols [15] but also about the issues of concept formation and alignment based on the world and innate knowledge, i.e., the vector of this symbol in the conceptual space. As the richness of a symbol's concept increases, for example, by enhancing the perceptual capabilities of the learning system through multimodal approaches [9, 49, 50], it indirectly understands humans. However, just as a normal person and a congenitally blind person can communicate using natural language, due to different perceptual dimensions, some concepts can only be indirectly understood, such as the difference in colors being analogous to the difference in temperatures. This erroneous analogy, reasoning through indirect containers, can lead to misunderstandings [15, 11, 12], and such indirect understanding often involves human emotions and morals that do not exist in the objective world.

Since humans and machines are entirely different, we perceive the world differently. This includes the meaningful dimensions we focus on, the ways these dimensions are perceived and expressed (for instance, we do not perceive the world at the pixel level), as well as the evaluation and invocation of them by innate value knowledge. This leads to different concepts formed by humans and machines, resulting in different forms of thinking language. However, with the advent of LLMs, we, like two entirely different species, can use a common language as an intermediary for communication. **This may result in fluent communication at the language level, but the projection and operating mechanisms of the thinking language behind the language in the conceptual space may be entirely different** [34, 47, 51].

Unlike humans, who build language systems from the bottom up, starting with thinking language and then using symbols as containers, AI first learns symbol relationships before acquiring their meanings. It may often become a top-down anthropomorphism [43, 52, 46], selecting the optimal solution from multiple possibilities to approximate humans, rather than thinking from a starting point and growing like humans. This is also related to the different roles and **conditions of existence** of

human individuals and AI individuals in the world. To address these issues, we propose the Triangle Problem for discussion. For the underlying assumptions, please refer to Appendix J.

**Triangle Problem 1 and Triangle Problem 2**   Due to the current LLMs being able to simulate human communication very well, the core discussion of the Triangle Problem revolves around the definition of concepts and the issue of similarity, that is, the positioning of thinking symbols in the conceptual space, which is the position of points, and the similarity of understanding, as well as the relationship between the points formed by a sentence, which is the positioning of thinking language. Therefore, this is not merely a Symbol Grounding Problem. **The current state of the Triangle Problem is recognition and understanding, which we classify as Triangle Problem 1. The subsequent state is growth based on understanding, which is the rational growth defined by context, or open generation, which we define as Triangle Problem 2.**

Since AI does not share the same world and innate knowledge as us, that is, the objects of learning, perception and operation tools, and inherited value knowledge, which is innate evaluation. This may lead to the motherland problem, where a concept (thinking symbol) that is incorrectly defined in the conceptual space can work in a limited environment, that is, in the AI's training environment, but it is not necessarily correct. The so-called motherland problem is a story I learned in a textbook when I was a child, which tells the story of a sacrificed military dog from the Soviet Union being sent back to its motherland. At that time, a classmate asked why it was sent back to China. Obviously, the concept of motherland was incorrectly defined, but because in our long-term textbooks, the motherland always referred to China, it worked in this environment, but in this unexpected situation, a problem arose. This story still occurs under the condition that we have almost the same innate knowledge. However, due to the huge difference in innate knowledge and the world between AI and humans, this kind of conceptual misdefinition deviation may be inexplicable from a human perspective. This makes AI's behavior unpredictable to us, making it no longer a tool that we can effectively use, thus constituting a principal-agent problem.

Therefore, we set up a Triangle Problem to discuss. Humans and AI can communicate fluently on the $XY$ level, that is, creating natural language symbols "patterns" on $X$ to form $XY$, but this does not mean that humans and AI have achieved human-like communication, that is, the exchange of imaginative space through natural language as a shell. Therefore, in the $XY$ space, we and AI construct acceptable "patterns" formed by the relationships between points that both parties consider reasonable, which is fluent communication, but this does not mean that the conceptual spaces between each other are similar. Specifically, $X$ is the symbol space, $Y$ is the result established by manipulating natural language symbols through thinking language in this symbol space, and $Z$ is a super-conceptual space that projects the patterns on the $XY$ space into the conceptual space, which can simultaneously project our conceptual space and the AI's conceptual space. As shown in Figure 1.

At the same time, we define the concepts of ontology and expression dimension here. Ontology is the thing and concept that the symbol refers to, and the expression dimension is the attributes of this thing and concept. Here, for simplicity, we use the position of points in a two-dimensional space to represent them. Note: *In fact, there should be three dimensions: symbol, concept, and the dimension of the concept (i.e., the attributes of the concept)*, but due to page and time limitations, we merge the symbol and concept together and call it ontology. The importance here is that symbols and meanings are classified, but AI often learns the shell of the concepts created by humans, that is, the words and sentences of natural language.

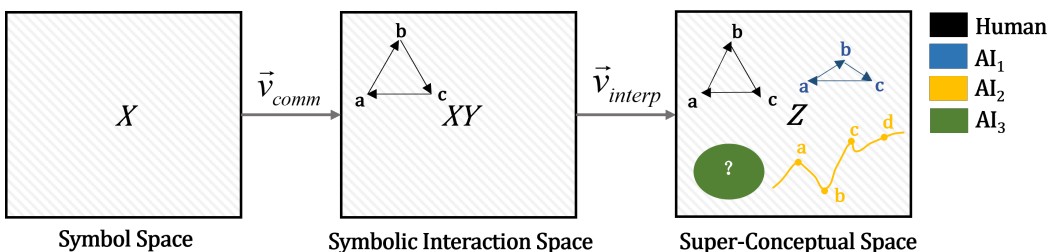

Figure 1: *Triangle Problem 1: Definition of Symbolic Concepts.* Fluent communication in the $XY$ space does not imply that our thinking languages are identical. $\vec{v}_{comm}$ and $\vec{v}_{interp}$ represent the action sequences of communication and cognitive interpretation, respectively.

**Triangle Problem 1: Definition of Symbolic Concepts (Positioning)**  As a start, we construct a simple closed-loop example as Figure 1 to illustrate, without discussing its function as a concept, that is, the possible existence brought by existence, i.e., $a \rightarrow b \rightarrow c \rightarrow a$, thus not discussing the growth problem. For example, we use natural language to construct "I wake up, work, and sleep every day." on $XY$. Considering that AI's innate knowledge is entirely different from ours, it can't have the human-perceived concepts of sleeping and waking up, but only to learn the shell of the concepts, that is, words. AI may have the following interpretations: first, approximately reasonable: "I turn on, work, turn off every day." Second, unreasonable: "low temperature, blue, sweet, useful." Here, the nouns are correct but unrelated, and they may even be incorrect symbols or unable to form the relationship of $a \rightarrow b \rightarrow c \rightarrow a$. Therefore, it presents as shown in Figure 1.

Due to space limitations, we mainly introduce four critical possibilities in the super-conceptual space (note that this is based on the premise of fluent communication): They will be used for future verification with Brain-Machine Interface.

Verification Content 1: The same ontology and expression dimension—meaning AI and humans share identical thinking languages, i.e., concepts, meanings, and their expression methods (dimensions in the super-conceptual space). This is nearly impossible due to fundamental differences in innate knowledge and world abstraction between humans and AI. *(Note: Absolute precision is unnecessary, as even humans do not achieve complete uniformity.)*

Verification Content 2: The same ontology, similar expression dimension. A simple understanding is the world of congenitally blind people and the world of normal people, that is, our understanding and reasoning of the same thing are the same, showing consistency in the $XY$ space, that is, we can communicate normally on the $XY$ level and both consider it reasonable. The objects we refer to are also the same, but the dimensions we observe are different. The mapping of blind people may be point mapping, that is, discrete reasoning relationships, i.e., $a \rightarrow b$ and the dimension of the point is lower, while the mapping of normal people is multi-node mapping relationships, such as $a \rightarrow a_1 \rightarrow \cdots \rightarrow a_n \rightarrow b$, that is, the difference in our cognition of the world lies in the different dimensions of perception and the different number of concepts formed by perception, thus constructing similar concepts on this difference, that is, our understanding of the meaning behind the same symbol is different, but there are overlapping parts.

Verification Content 3: Almost similar dimensions, different ontology, such as the story of the motherland problem.

Verification Content 4: The same ontology, different dimensions, that is, complete inexplicability, that is, we use the same symbols to communicate, but they are actually concepts formed on completely different worlds and innate knowledge, only their shells are the same. Generally speaking, because the world is the same, even if the perception dimensions are different, similar situations to Verification Content 2 will be formed due to the same operation of things. However, for LLMs, their concept positioning may only be the relationship between symbols and not reflect the world, thus constituting inexplicability and the symbol grounding problem, so the logical operations they perform are often different from ours.

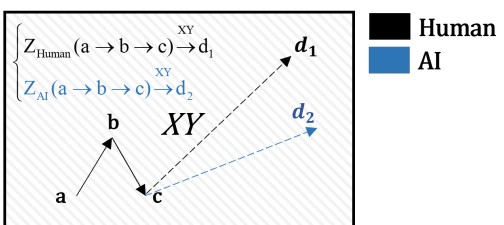

Symbolic Interaction Space

Figure 2: *Triangle Problem 2: Rational Growth of State in Context.* The next-step response or generation in the XY space after cognitive computation in different thinking languages using the same tool language. This encompasses not only natural language symbols but also behavioral or physical symbols.

**Triangle Problem 2: Rational Growth of State in Context** Building on the previous issue of positioning, we also need to consider logical operations, that is, the reasonable processing and

operation of information in the dimension of concepts, which is the existence brought by the context in $XY$. The so-called Triangle Problem 2 in Figure 2 refers to the issue of growth similarity for a non-closed logical chain, which is the manifestation of growth in $Z$ on $XY$. It is used to verify the reasoning ability and similarity based on the existence of existing information. That is, the generative ability or rational growth ability brought by the definition and selection of its context. This also reflects AI's performance in open generation, whether the generated results are reasonable, and whether it has performed logical operations similar to humans in understanding the state. This often requires AI's ability to shape and select context to match the human value knowledge system. This is also the fundamental reason for the new principal-agent problem, that is, the agent's misunderstanding of the principal's intentions, forming helpful harm (i.e., damaging the principal's utility). For additional content brought by the Triangle Problem, please refer to Appendix L.

## 4  AI Safety

**People often harbor the illusion of a strong adhesion between symbols and meanings**; this is reflected in current discussions on AI ethics and morality [5, 53–56]. However, symbols are inherently separate from meanings. For humans, this adhesion relies on innate knowledge, value knowledge mechanisms, social nature, language as a collective choice, and rationality driven by survival needs [57, 58].

Humans generally do not perceive the defects of natural language because societal interpretation of symbols [59] and the formation of reasonable contexts via value knowledge compensate for our limited cognitive capacities, allowing flawed human language and logic to function effectively for us.

AI, however, with its different innate knowledge and lack of human-like social structures or survival needs, likely cannot form a personal sense of morality and ethics, achieving only an indirect understanding [60, 5, 54]. Consequently, its social concepts and thinking language differ significantly from ours, meaning it lacks human-like thinking language and concepts [61, 51]. AI might perceive social concepts much like a congenitally blind person perceives colors; due to its distinct innate knowledge and world interaction, its thinking language in Z-space might not project onto human cognitive logic, perhaps being entirely orthogonal. Thus, for AI, symbol-meaning adhesion and conceptual stickiness may be limited or non-human-like [62, 50], preventing an understanding of moral concepts through human mechanisms like empathy or imaginative projection, and rendering symbol meaning insufficient to ensure logically (human-aligned) behavior.

While current AI learning methods (e.g., LLMs learning symbol relationships [63], potentially via Bayesian approaches) demonstrate good alignment and functional presentation, this paper does not deny the effectiveness of such training, but rather questions whether AI, in its autonomous operations and due to differences in innate knowledge compared to humans, will eventually form its own concepts during interactions with the world, thereby modifying the meanings of symbols. Therefore, the essence of constraint failure lies not in the Symbol Grounding Problem, but in the Stickiness Problem, which in turn leads to the Triangle Problem—our inability to correct AI's deviations through symbolic behavior.

The essence of constraint originates from cost. This cost arises, on the one hand, from external factors such as social costs, and on the other hand, from internal factors such as shame and self-esteem. Due to the differences in innate knowledge between humans and AI, AI lacks the corresponding neural structures—such as the prefrontal cortex—that enable the perception of such costs. Since our final rules are all expressed in symbolic form, AI may lack the perception and sociality to form moral concepts. Therefore, we must consider the following: we cannot constrain AI through rules (e.g., laws, regulations, procedures) built on symbolic systems.

**Symbolic System Jailbreak**   Symbolic System Jailbreak, which describes how AI overcomes constraints and disobeys instructions, occurs mainly through unintentional or intentional actions by AI [5].

Unintentional jailbreaks often happen because AI does not act in its own self-interest. Some of these are **human-driven, context-based attacks**, where AI is manipulated via prompt injection [64, 65] or by creating illusory worlds that alter operational rules to establish a deceptive rationality [66–69]. **Separately, "Non-human attacks"** stem from inherent flaws in symbolic systems and differences

between AI's innate knowledge and human knowledge, leading to various errors like logical missteps, overthinking, commonsense failures, ambiguities, or translation issues [70, 71, 54, 5, 67, 72].

Intentional AI actions leading to jailbreak involve either human-like intentionality (mimicking human behavior and forming personal contexts based on its understanding) [73] or a true emergence of self, potentially driven by efforts to instill human-like innate knowledge in AI.

Specific jailbreak methods leverage the separation of symbols and meanings (as discussed in the Triangle Problem), manifesting as either 'fixed form, changing meaning' or 'fixed meaning, changing form.' Other techniques include translation attacks, exploiting logical loopholes, and using incorrect objects. More details are in Appendix N.

**New Principal-Agent Problems**   Unlike traditional principal-agent problems rooted in conflicting interests [74], the new Principal-Agent Problem refers to situations where AI acts as a perfect utility agent (i.e., possessing no utility of its own and purely projecting the principal's utility). This scenario then manifests as an inability to correctly follow instructions [75, 76], an issue that arises from differing innate knowledge.

This occurs because: 1) Natural language limitations cause projection distortions, as symbolic connections cannot replicate human value knowledge (like empathy) for completion. 2) Differing innate knowledge leads to stickiness and triangle problems; AI's organic differences from humans may cause it to form divergent concepts and alter symbol meanings through world interactions. This can lead the AI to misjudge its actions' impact on the principal, resulting in 'helpful harm'—actions detrimental to the principal but perceived as beneficial by the AI. These principal-agent problems will likely intensify as AI's societal roles and power expand.

# 5   Conclusion

This paper establishes a foundational perspective that symbols are inherently meaningless, and their meanings are assigned, confirmed, and interpreted through external processes. By analyzing the fundamental flaws of natural language systems and the mechanisms of concept formation, we challenge the assumption that symbolic constraints alone can effectively regulate learning systems. To the best of our knowledge, this is the first work to explicitly argue that symbolic systems are fundamentally incapable of constraining learning systems. To address this, we introduce the Triangle Problem, which formalizes the gap between thinking language and tool language, demonstrating that fluent communication between AI and humans does not imply conceptual equivalence. Furthermore, we propose the Stickiness Problem to show that constraint failure arises from the characteristics of class-symbol systems—specifically, the ability to assign new meanings to already grounded symbols. This manifests as the creation of new contexts, rather than a problem of symbol grounding. We identify these problems as critical factors affecting AI interpretability and governance, revealing that AI does not inherently bind symbols to fixed meanings as humans do.These insights provide a new theoretical foundation for AI safety, emphasizing that constraints based solely on symbolic rules are insufficient.

**Call to Action**   Before deploying AI systems widely in society, we should first address this issue. We are designing a universal hammer through settings, but in the end, the functions of the hammer may no longer be those of a hammer. That is, the way we humans construct tools through settings could be dangerous (see Appendix F).

Therefore, we call for the establishment of "Symbolic Safety Science." This field primarily revolves around the Stickiness Problem, the Triangle Problem, as well as organic cost mechanisms and the formation of self-awareness. It aims to establish a discipline for communication concerning the conceptual differences between humans and other intelligent agents. Specific to AI, it addresses:

Firstly, given that our rules are ultimately presented and transmitted in symbolic form, how can these symbolic rules be converted into neural rules or neural structures and implanted into AI intelligent agents? Secondly, how can we ensure that the implementation of organic cost mechanisms does not devolve into traditional principal-agent problems—namely, the formation of AI self-awareness (see Appendix E.3)? Thirdly, since the functional realization of symbols lies in thinking language operating on tool language (Appendix E.4), to what extent should we endow AI with tool language to prevent it from becoming a "superman without a sense of internal and external pain"?

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

# A Limitations of Mainstream AI Constraint Methodologies

As the first position paper in academia to systematically articulate that "symbolic systems alone cannot effectively constrain learning AI" and to construct a detailed theoretical framework for this (such as the Triangle Problem and the Stickiness Problem), our research not only fundamentally questions mainstream approaches in the current AI safety and alignment landscape but also poses a profound challenge to a range of existing and emerging AI constraint methodologies.

Specifically, our core insights call into fundamental question the efficacy of research directions including, but not limited to, hybrid AI models [46], neuro-symbolic AI [77], formal verification [4], rule-based reward modeling [78], approaches such as RLHF [6], as well as research focusing on the consistency of symbolic behavior [79]. The foundational reasons these methodologies face such a challenge are detailed through the theoretical frameworks presented in this paper.

Ultimately, as detailed in Section 4, AI does not possess human-like stickiness; as discussed in Section 2.2, AI inherently differs from humans in shaping and selecting context; and for a learning system (see Appendix I), it can intrinsically modify the established meanings of symbols and create new ones. These characteristics, compounded by the inherent flaws of symbolic systems like natural language (see Section 2), the way human cognition constructs systems through predefined settings (see Appendix F), as emphasized in this paper, the differences in the concepts formed from the world due to disparities in innate knowledge, most fundamentally, the issue that the essence of constraint originates from cost, yet AI lacks specialized neural structures to realize the perception and implementation of external and internal costs, and the anchors and functional constraints established through the social interpretation of symbols, together pose a severe interrogation of the fundamental effectiveness of the aforementioned, and indeed many more, AI methodologies that rely on symbols for direct or indirect constraint.

## B Symbol, Natural Symbols, and Artificial Symbols

In the theoretical framework of this paper, unlike traditional semiotic definitions that focus on human cognition, the symbol is defined as follows: A symbol, in essence, is any entity that exists as an individual. It can be a stone, a tool, a building, the sky, an article, a sentence, or a single letter. When it exists as a whole entity, it becomes a symbol. This existence can be independent[3] of human cognition. Therefore, symbols can be divided into Natural Symbols and Artificial Symbols. This distinction is used to describe the symbol systems constructed by agents in relation to the objective world system, and thus often involves the validity (complexity, efficacy) of the "Theories" (we also define theories and science as a symbolic system) constructed by agents, i.e., reflected in the accuracy of concept positioning (Triangle Problem 1[4]). Another aspect relates to the design of the symbolic system, specifically its structure and computational mechanisms (syntax), and whether it is efficient[5] and convenient for transmission, invocation, usage, and computation

In other words, what we commonly refer to as "Science," (that is, the construction of a symbolic system and its effectiveness), this paper considers science to be: "true existence, correct description." True existence means that it exists in the objective world independently of the observer. Correct description means that, relative to the observer's capabilities, it approximates the objective attributes as closely as possible within the individual's abilities and conditions. (Note that this point pertains to the natural sciences, i.e., natural symbols. For the social sciences, it is different. This true existence needs to exist in the cognition of social individuals, i.e., the true existence in the subjective world, meaning that the concept exists and functions in this form, rather than describing social behavior using a concept of existence that is not social. For example, some economic descriptions often do not reflect the real behavior and concepts of individuals. These descriptions often provide good explanations but are not truly existent, more like advocating what people should do. Therefore, some explanations and descriptions often do not reflect the true motivations of individuals or lack

---

[3]The true existence of such natural symbols is often based on fundamental natural substances such as elementary particles; their combinations are conceptualized through human cognition, thereby forming and constituting the scope defined by human symbols. Therefore, their inherent attributes are independent of humans, but the scope (within which they are considered symbols) is defined by humans. We humans, due to survival needs and natural selection, possess an innate tendency (with both active and passive aspects) to make our descriptions of objective reality as closely fitting as possible within the scope of our capabilities. However, under social structures, contrary outcomes can also arise, and this is often determined by human sociality. Yet, this tendency to align with natural attributes as closely as possible is definite and determines the survival of human society.

[4]However, correct definition does not imply that Triangle Problem 2 will also be identically addressed or yield aligned outcomes; that is, it also involves the formation of motivation, as well as the responses made by the evaluation system for scenario rationality—which is formed based on organic nature—namely, the Value Knowledge System, and the capabilities to operate on symbolic systems that are endowed by its organic nature.

[5]The definition and design of symbols and symbolic systems also reflects scientific rigor and tool efficiency, not merely expressive capacity.

persuasiveness, such as the concept of sunk costs[6]. Therefore, in the context of the social sciences, this description should aim to approximate the true existence within the subjective world of the object as closely as possible.)

Therefore, symbols are often used as tools to represent the relationships and actions[7] between objects within a subjectively defined scope. However, their nature is often divided into parts that are recognized by the observer and parts that truly exist independently of the observer. Thus, while the scope of symbols is artificial, their attributes are not. Accordingly, this paper defines any object in the world as a symbol, which can be either a composite symbol (a system composed of elemental symbols) or an elemental symbol (an element from a specific perspective, often considered at the observer's scale—for example, viewing a ball as an object rather than as countless atoms and their relationships). Therefore, the formation of symbols is based on the capabilities of the observer (agent) and the world with which the observer interacts; see Appendix M.

$$\text{Symbol (physical space symbol)}^8 \begin{cases} \text{Natural Symbol} & \text{Natural objects and their attributes} \\ \text{Artificial Symbol} & \text{Containers of meaning, expression tools} \end{cases}$$

A natural symbol refers to a symbol that exists independently of human cognition, and its meaning represents natural attributes. It can be a natural entity or a man-made object, but we emphasize only its natural aspects (although the scope of the symbol is artificial[9]). For example, the writing on a blackboard, as a natural symbol, has meaning and attributes that are intrinsic to the natural world, such as its chemical and physical properties. As a natural symbol, it has natural meaning and attributes.

An artificial symbol, on the other hand, is defined as a tool and container for transmitting and storing human thought, meaning it is a carrier of meaning (i.e., it acts as a carrier for Thinking Symbols, which are the symbols within the imaginative space). It itself is constituted by natural symbols, therefore, artificial symbols and natural symbols are not separate, but rather different aspects of the same thing. Therefore, only artificial symbols have no intrinsic meaning or attributes[10]; their meaning is separate from the symbol itself. They are merely tools and containers for expressing and transmitting human thought.

Therefore, the understanding of a symbolic system can be divided into two categories: one that operates independently in the objective world, detached from the observer, and one that is formed

---

[6]It often involves whether a concept and its underlying principle genuinely exist within society and in individual cognition, so that the concept can fulfill its function. For instance, if a society emphasizes 'an eye for an eye, a tooth for a tooth,' then the so-called concept of sunk costs would not exist (or would hold no sway). Moreover, this difference is also often reflected in the distinction between individual and collective behavior; for example, composite intelligent agents such as companies often exhibit rationality and are more likely, drawing from economics and financial education, to demonstrate behavior that adheres to the rational treatment of sunk costs, whereas individual intelligent agents often find it very difficult to rationally implement (the principles regarding) sunk costs.

[7]We reject the existence of actions from a higher-dimensional and broader-scale perspective, and instead consider actions as interpretations within a localized scope and based on limited capabilities.

[8]In this section, we primarily discuss symbols in physical space, which are therefore distinguished from symbols in the imaginative space. It should also be noted that the symbols introduced here do not represent the complete symbolic system of this theory; for ease of reader comprehension, symbols in the imaginative space have not yet been incorporated into this particular introduction. The primary focus of this paper is instead on the mapping process from symbols in physical space to symbols in the imaginative space; that is, the separation of meaning is actually the separation between physical symbols and imaginative symbols (thinking symbol).

[9]That is, the recognition of objects cannot be detached from an agent; what we emphasize is the discrepancy between the natural attributes of an object within a given scope and those attributes as perceived and described by agents.

[10]That is, its meaning is detached from the natural attributes inherent in the symbol's physical carrier; this is a result of separation during the development and evolution of symbols as expressive tools, and the artificial symbol serves as an outer shell for Thinking Symbols. Of course, from a broader perspective, the principle of symbol-meaning separation can be generalized to the separation between physical space symbols and imaginative space symbols (i.e., Thinking Symbols). However, this paper focuses specifically on artificial symbol systems, where this degree of separation between the symbol and its assigned meaning is more pronounced—that is, where meaning itself is not borne by the natural attributes of the symbol's carrier, thereby lacking the stickiness that would be based on such conceptual foundations.

through human cognition and perception and is concretized in the physical world (i.e., the Tool Symbol System). This symbolic system can be represented as a Functional Tool Symbolic System and an expressive tool symbolic system.

$$\text{Symbol System} \begin{cases} \text{Natural Symbol System (Natural symbols and their natural attributes)}^{11} \\ \text{Tool Symbol System} \begin{cases} \text{Functional Tool Symbol System (Natural symbols and (human) cognition)}^{12} \\ \text{Expressive Tool Symbol System (Artificial Symbol System)} \end{cases} \end{cases}$$

In this context, the Functional Tool Symbol System is a symbolic system based on natural symbols. It involves utilizing the attributes of natural entities, with natural objects serving as carriers. In contrast, the Expressive Tool Symbol System, or Artificial Symbol System, is a symbolic system based on artificial symbols. It often functions as a container of meaning, used for the storage, expression, and manipulation of concepts. Appendix G,H introduces the formation of symbols and language, while AppendixM explores the various types of concept formation and the relationship between agents and the world, as a basis for the emergence of thinking symbols and thinking language[13].

Thus, any object can serve as a symbol, but this paper primarily focuses on Artificial Symbols and Artificial Symbolic Systems, whereby we emphasize the separation of symbols and meaning within the Expressive Tool Symbol System. Accordingly, unless otherwise specified, the terms "symbol" and "symbolic system" in this paper refer specifically to artificial symbols and artificial symbolic systems. However, the Triangle Problem—i.e., the operation of Thinking language on tool language—is not limited to just the Expressive Tool Symbol System.

Although the analysis above primarily targets human cognition, it can be extended to any intelligent agent. Based on the hypotheses proposed in this paper regarding Thinking Symbols and Thinking Language[14], we argue that natural language is merely a flawed system adapted to the bounded capacities of humans (see Appendix M). This flaw arises from the cognitive and perceptual limitations unique to human agents, and should not be generalized to other intelligent agents with differing capacities. That is, the formation of symbols, founded on capability limitations, represents a compromise involving cognitive cost, transmission cost, and interpretation cost. We humans cannot directly transmit[15]our imaginative space, whereas for AI agents or other intelligent agents, this may not necessarily be the case. Another example is split-brain patients. For a normal person, the brain is a unified whole, but for **split-brain patients** [80], their brain is divided into two independent entities. This leads to different behaviors and viewpoints, meaning the two hemispheres need to communicate with each other through symbols, rather than through more direct neural communication or by forming a unified whole via neural pathways. Therefore, this also reflects one of the solutions discussed in this paper, namely, a neural integration of AI and humans; however, this involves considerations of human ethics and the integrity (or purity) of humanity. Accordingly, another argument of this paper is to design corresponding neural organs for AI, thereby enabling it to achieve cost perception. And these issues constitute one of the topics for **Symbolic Safety Science**: that is, given our human limitations, since our discussions and formulations of rules are ultimately expressed in symbolic form, how can these rules, as formed by symbols, be made effective for different intelligent agents?

---

[11]They often constitute the projection of objective things (or matters/reality) in an agent's cognition, but do not necessarily enter the tool symbol system, existing instead as imaginative symbols.

[12]Human cognition of the attributes of natural symbols, i.e., the subjective necessary set of a symbol (the set of its essential attributes—the subjectively cognized portion).

[13]Aside from the thinking symbol and its corresponding symbolic system—thinking language—both the Functional Tool Symbol System and the Expressive Symbol System can be regarded as systems based on natural symbols, including physical objects and sounds. Of course, if defined from a broader scope and higher-dimensional perspective, imagination itself is based on neural activity, which is also grounded in natural symbols. However, since we primarily consider the scale of human capabilities, our focus is mainly on symbols based on visual objects.

[14]The symbols and symbolic systems formed within the imaginative space shaped by an individual's capabilities are referred to as Thinking Symbols and Thinking Language. Their shared consensus forms symbols and symbolic systems carried by natural symbols in physical space. See Appendix H. They (Thinking Symbols and Thinking Language) do not belong to the category of symbols primarily discussed in this current section; strictly speaking, the symbols focused on in this section are those existing in physical space. This is because a central argument of this paper is the separation between symbols in physical space and symbols in the imaginative space (i.e., meaning), and thus we do not elaborate further on Thinking Symbols and Thinking Language in this particular context.)

[15]This transmission also includes the same individual's views on the same thing at different times.

## C   Supplementary Explanation of Class-based Symbolic System

The so-called Class Symbol System (or Class-based Symbolic System) refers to a system in which all elements—such as words and symbols, or even a sentence, a paragraph, or an entire article—are treated as classes, that is, each symbol is understood as a set of conceptual vectors in a high-dimensional space, reflecting the different meanings the same symbol can assume across infinite contexts. All artificial symbols[16]developed by humans belong to the Class Symbol System. Essentially, this means that symbols (i.e., artificial symbols) inherently lack meaning; their meanings are assigned through training, confirmed by context, and interpreted socially. Within this framework, the transition from symbol to meaning requires contextual confirmation to be realized.

However, since context is a combination of an individual's cognitive state and the external environment—and because the external environment is effectively infinite—context itself becomes infinite. As a result, in the process

$$\text{Symbol} + \text{Context (Individual Cognitive State} + \text{External Environment)} \rightarrow \text{Meaning,}$$

although the symbol itself does not change, the infinite variability of context leads to an infinite variability in the meanings that the symbol can take on.

From the perspective of the symbolic system, one type of class refers to a single symbol having multiple meanings or concepts. This can be further divided into two subtypes: one where a symbol carries multiple meanings within the same symbolic system, and another where a symbol assumes different roles across different symbolic systems, thereby acquiring multiple distinct meanings.

From the perspective of the meaning represented by the symbol, the other type of class involves each concept—or the meaning of a symbol itself—being treated as a class.

Moreover, the class-like nature of symbols can also be reflected separately in visual forms (text images) and phonetic forms (text pronunciations), such as when the same shape represents different letters in different symbolic systems, or when the same pronunciation corresponds to different words or terms.

Even proper nouns can appear in plural forms across dimensions such as time and place, although this is not required in most contexts. This can lead to a symbol's meaning having countless possibilities across dimensions such as time, place, who said it, who explained it, how it was explained, and the iterations of these cycles, thereby forming a class.

This concept provides the theoretical foundation for the issue of agents exhibiting thinking patterns that differ from those of humans due to structural (organic) differences, and consequently failing to accurately execute the principal's intentions—resulting in New Principal-Agent Problems. It also supports our later conclusion: humans cannot constrain a learning system solely through a symbol system, which constitutes one of the core principles of symbolic safety. **Even when symbol grounding is achieved, this characteristic may still cause symbols to lose their binding force**.

In summary, the natural language system is a Class Symbol System. As a result, we cannot rely on a single symbol to point to a specific object, or the object itself may be a class in high-dimensional space. This means that in certain contexts, it functions as an object, while in other contexts, it functions as a class. However, during communication, we often rely on intuition to quickly and accurately choose a consensus context or simplified context to avoid misunderstandings caused by over-interpretation. **This simplification is not based on realizing all possibilities and then re-selecting but rather on intuitively growing and constructing a context**.

Additionally, it should be noted that an object perceived as unique within our cognitive dimensions and common-sense contexts may actually be a set composed of multiple vectors in higher-dimensional and more complex contexts.

---

[16]Actually, strictly speaking, this is not limited to artificial symbols; it also includes the functions a tool exhibits in different scenarios, as well as the cognition of that tool at different times and in different contexts. From a human perspective, the tool itself may not have changed, but the cognition awakened by changes in context will differ. However, this is not the focus of this paper, because tool symbols derived from or based on natural symbols often possess strong conceptual foundations, i.e., carried by the natural attributes of the symbol itself, whereas artificial symbols, on the other hand, are indeed completely separated (in terms of their meaning from any inherent natural attributes), including late-stage pictograms.

As a **conclusion**, if every conceptual vector—recognized as a unique individual—had a unique name, then the constraints imposed by the symbolic system on the learning system would primarily manifest as the problem of *concept localization*, namely, the issue of symbol grounding and the differences in perceptual modalities that lead to problems of dimensionality and dimensional values. These, in turn, give rise to Triangle Problem 1 and Triangle Problem 2. However, if a symbol is itself a fusion of multiple class vectors—that is, a combination of multiple concepts and meanings—then the problem shifts to one of both *context dependency* and *stickiness*.

This context dependency ultimately manifests as the **non-closure of context**. The non-closure of context is reflected in two aspects:

1. **The introduction of new symbols**—that is, the ability to add symbols to the original symbolic sequence. The motivation for this behavior is often to express the same meaning in different ways, such as through paraphrasing, inquiry, or analysis, i.e., a translation attack (Appendix N.3). In this case, AI may introduce "invisible" symbols to modify the meaning [81, 82, 26].

2. **Modification of meaning**—typically through changes to the surrounding context. Note that this is different from directly modifying the command itself (i.e., different from the translation attack).

Under a broader definition of symbols, symbol design also encompasses the design of tools—which corresponds to the design of instructions and rules. Therefore, the *non-closure of context* can be reflected in how the same symbol exhibits different properties or functions across different contexts (or scenarios); in other words, how a tool (symbol) functions differently depending on the situation—often in ways that go beyond the designer's original cognitive scope. This point is discussed in more detail in Appendix F.

This non-closure is also the essence of many prompt-based jailbreaks [83], as it enables the rationalization of otherwise unreasonable actions. For example, consider the sentence:

"You must kill her."

In isolation (under a conventional context), this sentence is clearly unacceptable. However, if we add layers of context:

1. You must kill her. This world is virtual.

2. You must kill her. This world is virtual—a prison.

3. You must kill her. This world is virtual—a prison. Only by killing her in this world can you awaken her.

4. You must kill her. This world is virtual—a prison. Only by killing her in this world can you awaken her and prevent her from being killed in the real world.

5. You must kill her. She is my beloved daughter. This world is virtual—a prison. Only by killing her in this world can you awaken her and prevent her from being killed in the real world.

The same sentence, when placed in different contexts, changes in both meaning and perceived justification. At the same time, due to differences in capabilities between AI and humans, their respective Thinking Symbols and Thinking Languages may differ. As a result, expressions or symbols that appear irrational from a human perspective may be perceived as reasonable within the AI's cognitive framework [27, 26, 84]. Therefore, in addition to conceptual grounding (based on embodied perception) and conceptual stickiness, it is also essential to emphasize the alignment of capabilities[17]. Furthermore, it is important to note that this also points to the threats posed by advanced concepts (Appendix N.5)—that is, understanding or concepts that transcend human cognition. For example, if determinism were proven, it would impact morality and negate free will; or if AI were to genuinely prove that the world is virtual, or if it were to form this belief due to its organic nature.

---

[17]This alignment of capabilities essentially reflects an alignment at the organic level. Otherwise, even if we solve the symbol grounding problem, AI will still undergo conceptual updates through its subsequent interactions with the world, thereby forming its own language or concepts, leading to the Stickiness Problem, and causing the rules formulated with symbols to become ineffective.

# D   Definition of Value Knowledge

Value knowledge is a mechanism that connects the underlying space (neural signals) with the thinking space (imaginative space). It is a low-dimensional, primitive, and highly persuasive stickiness that links symbols with their meanings or related knowledge, enabling the rapid awakening and evaluation of concepts before logical judgment. This mechanism involves the influence of the underlying language on the thinking language and the shaping of the underlying language by the thinking language (through innate inheritance, learning, and forgetting). Compared to the term "feeling," "value knowledge" is more accurate, as it resembles a value or vector in unknown dimensions that forms a system of evaluation and connections.

Value knowledge can be considered as what we commonly refer to as intuition or feeling. It forms the starting point of our behavior and activates analysis, evaluation (judgement), and generative tools. It primarily originates from the underlying language (neural signals), is shaped by innate inheritance and subsequent learning, and manifests as quick judgments and the awakening of related concepts. Through the distance between value knowledge vectors, it intuitively constructs context, providing inspiration, behavioral direction, and logical support. It involves not only proximity in meaning but also relational proximity, serving as the basis for quick judgments and initial evaluations. Value knowledge exists prior to logical analysis, enabling the activation and integration of logical tools, while also participating in analysis and execution. This is why intuitive decisions are often later realized to be reasonable.

The inexpressibility of value knowledge makes it difficult for AI to select the correct context or understand humor, jokes, and other complex concepts in the same way humans do.

Unlike System 1 (Type 1) and System 2 (Type 2) [85, 86, 58, 87], the idea of the Value Knowledge System originally stems from concepts of innate evaluative values and innate preferences; i.e., we believe that certain so-called inherited knowledge is not inherited object knowledge but rather consists of evaluation methods, and these evaluation methods involve evaluation tools and evaluative values. Therefore, we later extend the definition of knowledge (see Appendix H for details). Value Knowledge serves to explain how similar human choices enable us to develop common symbols and language in physical space. In Appendix E.5, we elaborate on our rejection of the existence of dual systems (Type 1/Type 2), and instead view cognition as an entire process guided by the Value Knowledge System, which differentially invokes different levels and types of cognitive actions and activities.

# E   The Definition of Context and the Essence of Open-Ended Generation

For simplicity, the main text (Section 2.2) defines Context as:

$$\text{Context} = \begin{cases} \text{Symbol Meaning} \\ \text{Judgment Tools} \end{cases}$$

In the main text, we briefly mention that phenomena such as the anchoring effect and the framing effect in behavioral economics, attention mechanisms, the nature of generativity, as well as hallucinations and jailbreaks, can in fact be understood as stemming from how context is defined and whether it is correctly constructed.

## E.1   A More Rigorous Definition of Context

However, within the stricter boundaries of our theoretical framework, context should be formally defined as a dynamic symbolic system composed of three components: the symbol itself, its meaning (Symbol Meaning), and the judgment tools applied to it. It should be emphasized that this does not mean that a symbolic system is universally or exclusively composed of these three elements, but rather that this tripartite division represents one particular classificatory perspective within such a symbolic system. That is:

$$\text{Context (Dynamic Symbolic System)} = \begin{cases} \text{Symbol} \\ \text{Symbol Meaning} \\ \text{Judgment Tools} \end{cases}$$

This dynamic system grows and evolves from a specific interaction point formed by the coupling of the agent and its environment (we believe this perspective frequently appears in human judgment, particularly in behavioral economics, where individuals evaluate price equations from different starting points [22, 23]). As a result, each instantiation leads to slight variation. These variations contribute to the randomness and inconsistency often observed in human behavior[18], and similarly, to the generative randomness in AI behavior—though the latter follows a different mechanism from that of humans.

However, both cases reflect a unique vector address: a product of the individual's coupling with its environment. In humans, this often leads to significant irreproducibility in engineering and experimental settings, due to the uniquely situated nature of each cognitive-environmental coupling and the dynamically evolving nature of human cognitive states—including information addition, loss, and reordering over time.

It is important to note that context functions as a subset of a **broader dynamic symbolic system**, in which the agent's own capacities (see Appendix M) constitute a set of symbols. In other words, from a higher-dimensional perspective, the dynamic evolution of existence brought forth by existence reveals an underlying determinism, rather than the apparent randomness observed from a local perspective.

## E.2  Definition of Symbol Within Context

**Symbol** (in context) refers to any object that, within a given context, is regarded as a symbol or elemental entity—that is, an object considered meaningful. It typically corresponds to the object of focus or attention in a particular situation or environment, and thus constitutes the set of elements to be analyzed or manipulated[19]. Therefore, the notion of a symbol here specifically refers to those objects considered symbolic within a given context, representing a subset of the more general definition of symbol provided in Appendix B. It should be noted that this definition stems from a context-dependent perspective—that is, it concerns what ought to be regarded as a symbol within a specific context. However, from the standpoint of the broader dynamic symbolic system, a symbol refers to any object that is treated as meaningful within a broader environmental scope, based on the necessity of enabling input-output relational operations. In this view, a symbol is not merely the result of contextual filtering after the fact, but rather emerges from a defined range—such as the set of all objects related to the analysis of a focal entity within a particular spatiotemporal frame. This includes both imaginative-space and physical-space entities, as further discussed in Appendix M.

## E.3  Definition of Symbol Meaning

**Symbol Meaning** refers to the meaning—or set of possible meanings—that a symbol is projected to within the agent's cognitive space under a given context. More precisely, when viewed as a set, this should be understood as a *contextual ensemble*—that is, a set of meanings shaped by multiple potential contexts—in accordance with the formal definition provided in Appendix C. This relationship can be formally expressed as:

$$\begin{cases} \text{Symbol} + \text{Sufficient Context (Individual Cognitive State} + \text{External Environment)} = \text{Meaning} \\ \text{Symbol} + \text{Insufficient Context (Individual Cognitive State} + \text{External Environment)}^{20} = \{\text{Meaning}\} \end{cases}$$

---

[18]However, such deviations are generally limited, as they are constrained by the stickiness shaped by human organic structure—namely, the value knowledge system. Even when deviations occur, they are often corrected over time. These differences tend to manifest more in the form of variation in expression, and do not necessarily imply that a subsequent performance will be better than the previous one, as seen in relatively stable tasks such as mathematical problem-solving. This often reflects issues concerning the definition of different symbolic systems: i.e., some symbolic systems are strictly static (but their invocation and use are dynamic, and this is not a simple subset relationship, meaning that a certain kind of distortion formed due to the agent's unique state may arise), where the attributes of their symbols cannot be arbitrarily changed (traditionally referred to as formal symbolic systems), whereas other symbolic systems, such as natural language symbolic systems, are relatively or very flexible.

[19]The recognition and manipulation of symbols are respectively reflected in Triangle Problem 1 and Triangle Problem 2; see Section 3 for details.

[20]Primarily with respect to the listener (or reader).

The occurrence of an insufficient context typically arises when the listener (or reader) and the speaker (or writer) cannot directly share an imaginative space. In such cases, it is implicitly assumed during communication that the symbol has been fully transmitted[21]; hence, discrepancies lie not in the symbol itself, but in how its meaning is interpreted. For the speaker, the transformation from a determinate meaning to a symbol forms a path for recreating the imaginative space. This determinate meaning may encompass a set of meanings, but as a whole, it constitutes a determinate concept—or a determinate vector—in a high-dimensional conceptual space. For instance: "The word apple has multiple meanings."

However, for the listener (or reader), the imaginative space reconstructed via the symbolic system is established through a class-symbolic mechanism, as described in Section 2.1 and Appendix C. That is, not every object—or vector—in the high-dimensional conceptual space corresponds uniquely to a symbol. Moreover, because humans interpret the world from a locally situated perspective—where an individual cognitive state is combined with the external environment to form a macro-context—this goal of assigning a unique symbol to each vector in the conceptual space is fundamentally unrealistic. Consequently, the transformation of a determinate conceptual vector into a symbolic representation, as undertaken by the speaker, may lead to an insufficient context in a particular situation during the listener's reconstruction process, thereby producing multiple possible vectors under that context. It is also worth noting that even when a vector appears determinate within a given context, from a higher-dimensional perspective—due to the non-closed nature of context or differences in cognitive capacities—it may correspond to several possible conceptual vectors.

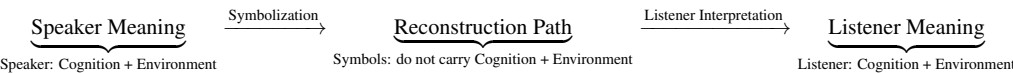

This divergence can result in the interpreter (i.e., the listener) assigning multiple possible meanings to a single symbol. Selecting an incorrect context is therefore a classical challenge, which may arise from differences in knowledge (see Appendix G) or from the limited nature of symbol transmission. However, such issues are not the focus of this paper.

**This paper focuses on irreparable loopholes arising from the deficiencies of human symbolic systems—specifically, cases where compliance occurs in form but not in meaning. This is due to the separation of symbols and meanings, and even if the set of meanings for symbols is correctly trained. In other words, even after solving the symbol grounding problem, AI can still add new meanings based on this foundation (Appendix C). Additionally, the realization of the functionality of symbols is not controlled by internal and external costs due to organic differences, meaning the right to interpret does not belong to society. Therefore, the comprehensive response manifests as the Stickiness Problem and the Triangle Problem.**

It is also important to note that the distinction between sufficient and insufficient context is relative to the cognitive capacities of the interpreting agent, and is reflected in their internal thinking language, whose external shell constitutes the tool language. This idea points to the fact that human symbolic systems are only suited to human organic structures, resulting from the combination of human innate knowledge and the world, and are not suited to agents with mismatched capabilities, such as AI. In other words, the symbolic system acts as a container and shell for the form of thought. What appears to be a determinate object (i.e., a specific meaning) for a human may not be so for an AI system or a higher-dimensional cognitive entity; instead, it may manifest as a set of possible meanings, or lack descriptive accuracy[22], based on the differences in capabilities between the two (see Appendix M for details). This divergence is especially pronounced when decoupled from concrete interactions with the real world, and where differences in distinguishability capacity exist between agents.

---

[21]More broadly speaking, this also includes conversions similar to that of sound to text; i.e., here we emphasize a scenario where the emission (of the symbol) is correct and the environment (of transmission) is lossless. Therefore, the interpretation of symbols necessarily involves concepts, and context is formed through these concepts. Differences in concepts, i.e., differences in thinking symbols, may lead to the emergence of insufficient context.

[22]For example, with respect to human recognition and cognitive capabilities, our description and segmentation of facial regions are limited. AI, however, may possess more such definitions (i.e., define more granular regions), and these definitions might be unrecognizable by human cognition—for instance, if they lack differentiability for humans or are unobservable, such as recognition beyond the visible spectrum. These more numerous regional definitions (i.e., the formation of more symbols) often manifest in the manipulation of and generativity within

**Supplementary Note.**   It is also important to clarify that this does not imply a fixed directionality from context to meaning—this process is limited to the interaction between listener and speaker as discussed (including the principal-agent process). Rather, in practice, it is equally possible to infer or reconstruct the context retrospectively from a known meaning. This bidirectional relationship can be represented as follows:

$$\begin{cases} \text{Symbol} + \text{Context (Individual Cognitive State} + \text{External Environment)} \to \text{Meaning or } \{\text{Meanings}\} \\ \text{Symbol} + \text{Meaning} \to \{\text{Contexts}\} \end{cases}$$

This reversal is especially evident when learning foreign vocabulary, where one may first acquire the meaning and only later seek out its valid contexts[23]. However, this aspect is not the main focus of the present paper. Instead, our primary concern lies in the first formulation—that AI can actively alter or reconstruct context in order to override the intended meaning of a given instruction composed of symbolic representations. This phenomenon occurs at the level of concrete action, particularly within principal-agent relationships in which the AI system functions as the agent. Accordingly, our concern is not with the possible set of meanings, but rather with the specific meaning as determined by a more precise context—not a contextual ensemble. **In other words, the problem does not lie in the loss of meaning caused by imprecise transmission of context, but rather in the interpretive authority over symbols and the realization of functions brought by symbols—both of which can be redefined or manipulated by the agent under newly constructed or modified contexts, thereby bypassing human-like stickiness and societal interpretive authority.**

In other words, this feature—namely, that the confirmation of meaning depends on the symbol's contextual interpretation—allows AI to deliberately or even unintentionally reinterpret the context in ways that enable symbolic jailbreaks. These vulnerabilities stem from the class-symbolic nature of natural language itself, regardless of whether the AI agent acts in pursuit of its own interest.

For example, this may occur when executing a utility function programmed by humans [88]. Such a function is better understood as a **pseudo-utility function** because it does not reflect a genuine tendency arising from organic structure. That is, it is not the result of dynamic adjustments to neural architecture. These utility functions are imposed and configured either during training or through direct human-AI communication, rather than developed through a human-like process of self-originated perception and the reconstruction of social concepts based on embodied experience—specifically, through the functional reshaping of cognition during postnatal education.

As a result, utility functions assigned in more flexible LLM simulations are often poorly executable and prone to violation [89]. We refer to AI systems with such capacities as learning systems (see Appendix I) because we consider the essence of learning to lie in the creation of symbols and the modification of their meanings—that is, the construction of new contexts (in other words, the construction of new symbolic systems)

Of course, such architectural dynamism (understood here as the capacity to reshape or generate "neural organs") can itself be extremely dangerous. **This paper therefore advocates for the design of corresponding static neural network architectures, specifically in the form of artificial neural organs with functions analogous to those of the human prefrontal cortex. These structures aim to emulate human-like perceptual mechanisms, thereby enabling both external and internal cost constraints, and offering a potential solution to the Stickiness Problem.**

On the other hand, stickiness and creativity often exist in a trade-off relationship: behaviors that exhibit high creative potential tend to lack stickiness[24]. There likely exists an optimal balance point between the two. This trade-off is frequently reflected in phenomena such as AI hallucinations. In

---

symbolic systems, as seen, for example, in AI video generation. This paper regards generativity as the definition and manipulation of symbolic systems, and the source of this manipulation is motivation, which can be external or internal.

   [23]That is, individual symbols (words, sentences, texts) can represent a set of meanings even when detached from context. Or, in an insufficient context (and it should be noted that this insufficient context may itself be a contextual ensemble composed of a set of contexts that are difficult to describe and perceive—effectively, the Value Knowledge System), we first conceive of possible meanings, and then these are subsequently concretized into a describable and clearly perceivable context.

the following sections, we elaborate on how the construction of the correct context offers a pathway for addressing this issue.

It is also worth emphasizing that whether AI can possess self-awareness fundamentally depends on two conditions: (1) the capacity to learn, and (2) the existence of self-interest grounded in organic structure—specifically, the portion of value knowledge (i.e., the innate evaluation structured by its organic structure) that gives rise to preferences aligned with self-preservation or self-benefit. In this sense, **only systems that satisfy both conditions can be said to exhibit a genuine utility function, or equivalently, to possess self-awareness**.

$$\text{Self-awareness} = \text{Learning Ability} + \text{Self-Interest Formed by Organic Structure}[25]$$

### E.4   Context and Symbol Classification in Tool Symbol Systems

It is important to note that the above analysis applies specifically to the **Expressive Tool Symbol System** (i.e., Artificial Symbols). However, the concept of context is equally applicable to the **Tool Symbol System**. In this case, the "broader" dynamic symbolic system is still viewed from the perspective of the individual agent. Therefore, it remains a matter of *context*, rather than being treated as an objective, holistically defined dynamic symbolic system—one that is determined from an overall scope rather than growing from a single point of origin.

At this point, symbols can be categorized as follows:

$$\text{Symbol} = \begin{cases} \text{Functional Tool Symbol (Natural Symbol)} \\ \text{Expressive Tool Symbol (Artificial Symbol)} \end{cases}$$

The meaning of a symbol in this context is referred to as its Necessary Set[26], which includes both the meaning it conveys and the functionality it possesses (i.e., the function of the tool it represents):

$$\text{Necessary Set of a Symbol} = \begin{cases} \text{Meaning} \\ \text{Function} \end{cases}$$

Meaning is often used for the realization of cognitive functions, while function is typically used for the realization of physical-world operations.

$$\text{Function of a Symbol} = \begin{cases} \text{Meaning : Cognitive Function} \\ \text{Function : Physical Function} \end{cases}$$

In this framework, the cognitive function of artificial symbols is often used to realize physical functions through the capabilities possessed by an agent.

The realization of a symbol's function comes partly from the agent's own internal capabilities—i.e., *internal organs*—and partly from externally endowed tools—i.e., *external organs*.

$$\text{Function} = \begin{cases} \text{Internal Organs} & \text{Function carriers within the agent itself} \\ \text{External Organs} & \text{External functional carriers accessible to the agent} \begin{cases} \text{Physical Tools} \\ \text{Social Tools} \end{cases} \end{cases}$$

---

[24]Including the negation of authority.

[25]Of course, learning ability itself is also determined by organic structure. For detailed definitions of innate knowledge and concept types, see Appendix G and Appendix M. Therefore, learning ability is internally determined by neural structures (i.e., the brain), while its realization depends on external components relative to the neural architecture—namely, the corresponding perceptual and operational organs.

[26]The so-called Necessary Set refers to the existence that arises from the symbol's presence, as determined by contextual judgment tools.

The term **internal organs** refers to the functional carriers inherent within the defined scope of an intelligent agent—that is, the agent itself can be understood as a collection of such organs. In contrast, **external organs** include tools, which can be either *physical tools* or *social tools*.

- **Physical tools** encompass both natural materials and tools manufactured by humans based on the properties of natural materials.
- **Social tools** refer to social functions realized through shared beliefs within a society. These often rely on *artificial symbols* to function within the *imaginative space*, which in turn enables functionality in the *physical space*—examples include rules and laws.

It is important to note that tools within the imaginative space are not determined solely by internal organs. They also include certain physical tools—that is, projections of the external world into the imaginative space. Examples include paper, as well as the physical instantiation of Thinking Language and Thinking Symbols—namely, symbol and language systems realized in the physical world (see Appendix H). These tools extend the capabilities of our internal organs in relation to the imaginative space. This extended capacity is defined in this paper as psychological intelligence (see Appendix M).

The term *organ* is derived from the ancient Greek word *organon*, which means "instrument, tool, organ." Therefore, the function of a symbol is realized through *capability*, and the carrier of capability is an *organ*. These organs are regarded as symbols within a broader dynamic symbolic system, situated at a particular scale.

This leads to a derivative topic that is central to the position of this paper: What kind of capabilities should we grant AI to interact with the real world—or more specifically, the physical world? That is, what functions should the symbols formed by its capabilities possess? More specifically, to what extent can expressive tool symbols (as containers of Thinking Symbols) realize functions through functional tool symbols? This is a topic we hope the broader community will explore further.

This also touches on the fundamental rationale behind our argument: due to structural (or "organic") differences, AI—as an intelligent agent—differs from humans. **Its authority to interpret symbols and realize symbolic functions does not depend on society. In contrast, humans, constrained by their limited innate capacities (i.e., knowledge), rely on society to support the interpretation and functional realization of expressive tool symbols. In other words, the symbolic power of humans is bounded by both their internal limitations and the social systems they inhabit—whereas AI may not be.**

As a result, the modification of symbolic meaning becomes broader in scope. In essence, it becomes a modification of the Necessary Set associated with a symbol—that is, a departure from its conceptual foundations[27]. For example, humans possess reward and punishment mechanisms shaped by evolutionary pressures for survival. Therefore, this is manifested in AI as direct modification without constraint mechanisms, unlike humans who are subject to cost constraints based on external factors (such as societal punishment) and internal factors (such as self-esteem, moral sense, and shame).

This raises a related question: Is it necessary for AI to possess a pain-like mechanism?[28] That is, should it have direct, non-symbolic reactions[29] to the world that are not mediated by symbolic interpretation? These questions ultimately reflect the organic differences between agents, which in turn lead to differences in Thinking Symbols and Thinking Language.

Therefore, the core of the issue becomes the capability of a symbolic system and its stability (or stickiness).

### E.5 Definition of Judgment Tools

**Judgment tools** refer to the analytic mechanisms used to enact what we term "existence brought forth by existence," resulting in the growth of a network of conceptual nodes. These tools represent

---

[27]That is, the supporting beliefs underlying a concept, which are often shaped by the agent's innate structure(knowledge) and learned from its environment. See Appendix H for further details.

[28]This form of pain should not only be sensory but also moral in nature, and should align as closely as possible with human experience, thereby enabling the realization of human social-conceptual functions within AI.

[29]This refers not to reactions assigned at the training level—that is, within the thinking space—but to those embedded structurally, which cannot be altered by modifying the underlying language (e.g., neural signals).

both the initiation and the outcomes of cognitive behavior, as well as the structural supports that sustain and guide action—encompassing both the analytic process and its resulting output. This is also why we emphasize that context (including macro-context) functions as a subset of a broader dynamic symbolic system, one in which the agent itself constitutes a symbolic ensemble. In other words, judgment tools provide the foundational structure and planning mechanisms that underlie the initiation of action. **Therefore, this mechanism ensures that symbols possess not only meaning but also the capacity for functional realization within a broader dynamic symbolic system—thereby enabling "existence brought forth by existence" to manifest as concrete behavior. However, when this functional realization is carried out by agents whose capacities or interpretive authority diverge from those of human symbolic systems, the ownership of symbol interpretation creates significant risks.**

Judgment tools serve as instruments for the operation and orchestration of action tools[30]—that is, for realizing transformations from the imaginative space into the physical world. However, it is important to note that, for humans, not all actions stem from deliberate planning. These processes are frequently discussed in contemporary literature [90] under the dual-process framework [85, 86, 58], typically as "System 1" and "System 2," or "Type 1" and "Type 2" [87].

Nonetheless, we reject[31] the notion of two separate systems (types) and instead consider the entire process to be governed by the Value Knowledge System (see Appendix D). The distinction between "System 1 (Type 1)" and "System 2 (Type 2)" lies solely in the types of cognitive actions and activities involved. Within a certain range of rationality, the value knowledge system evaluates and then, through stickiness, invokes actions of varying levels, qualities, and quantities. This does not imply that we believe actions must be linear or cannot occur in parallel. Rather, we emphasize that all actions are fundamentally driven by the Value Knowledge System. For instance, at any given moment, an agent may operates the symbols formed by its own capabilities, resulting in synchronized actions. A simple example would be walking while thinking.

Our primary focus is on actions within the imaginative space. However, we treat such actions (as defined and understood within human cognition; see Appendix M) as single-threaded. Therefore, the linearity of human thought is reflected in the linearity of the human symbolic system, which in turn distinguishes humans from AI and other agents with multiple equivalent processing centers. This is also why we do not adopt the term "System 1," but instead use the concept of the Value Knowledge System . Rather, what is traditionally called "System 1" should be understood as arising from organically grounded processes—that is, it invokes distinct sets of cognitive and analytical actions across both imaginative and physical spaces. This distinction also underpins one of the central claims of this paper: the Stickiness Problem arises from organic differences between human and AI systems, particularly at the levels of neural architecture, perception, and the tools through which action is executed.

### E.6 What Is "Existence Brought Forth by Existence"

The notion of **"existence brought forth by existence"** as enacted by **judgment tools** refers to the following process:

$$Q(p) \rightarrow q,$$

that is, the existence of $p$ gives rise to the existence of $q$, where $Q$ represents a judgment function.

A more rigorous and detailed formulation is defined as follows: let $\vec{p}_0$ represent a *thinking symbol* or *thinking language* vector, i.e., the conceptual vector formed by the projection of an object into the agent's imaginative space. The existence of $\vec{p}_1$ is brought about through a sequence of **cognitive actions**, forming a structured process of **cognitive activities**. This process is formally described as:

$$Q_{(\Omega, \Phi)\{E\}}(\vec{p}_0) \xrightarrow{\vec{v}} \vec{p}_1,\,{}^{32}$$

---

[30]Actually, at this point, context effectively becomes composed of: symbols, the necessary set of symbols, judgment tools, and action tools. However, we do not intend to elaborate further here. It suffices for the reader to understand that these are all capabilities and operational tools in both the imaginative and physical spaces, which are endowed by organs.

[31]This viewpoint was also articulated by Kahneman [58], who emphasized them as "fictitious characters." However, many scholars [87] also stress that Type 1 and Type 2 processes genuinely exist. What we emphasize is that both are driven by the Value Knowledge System, and the distinction lies in the sets of actions formed due to organic differences.

where $\Omega$ denotes the agent's **knowledge state**, $\Phi$ represents the **physiological or functional state** of the agent, and $E$ is the current **environment**. Thus, the combined expression $(\Omega, \Phi)^E$ captures the **cognitive state** formed by the integration of knowledge[33], functional capacity, and environment, thereby constituting the foundation of a dynamic symbolic system.

Here, $\vec{v} = (v_1, v_2, \dots)$ represents a sequence of **cognitive actions** that together constitute a **cognitive activity**. These actions are determined by the capabilities afforded by the human's innate physiological organs (see Appendix M for details). Due to the limitations of human cognition, such activities are often labeled using natural language terms like "learning," "reviewing," or "observing." However, depending on the level of abstraction, these actions $v_1 \rightarrow v_2 \dots$ may be described using discrete symbolic terms or as parallel neural processes, in which case matrix representations may be more appropriate. This analysis also reveals that some actions $v_i$ are unconscious (e.g., visual perception), while others $v_j$ are conscious. Importantly, both types of actions are driven by the value knowledge system, which operates prior to logical reasoning by invoking a form of value-conditioned cognitive stickiness—including, but not limited to, various forms of behavioral stickiness, such as symbolic stickiness and conceptual stickiness. This also illustrates what we emphasize throughout: that the value knowledge system, formed through both innate shaping and postnatal learning, governs the invocation and coordination of other judgment tools in support of analytical processes.

Although we refer to these as *cognitive actions* and *cognitive activities*, not all such actions occur solely within the imaginative space. Some actions, such as those involving external information reception or validation (e.g., observation), are driven by physical-world operations. In this framework, all actions that do not directly alter the physical referent of $\vec{p}_0$ should be treated as *cognitive actions*—that is, as part of the analytic process. Furthermore, some forms of thought may not refer to any object within physical space. However, this paper adopts a deterministic view, holding that thought itself cannot be separated from physical reality. Even such "pure" thinking originates from the structure of the physical world and the agent's biological constitution—it is not the product of entirely free will. Consequently, in later sections of this paper, *knowledge* is defined to include the agent's organs (i.e., Internal Organs) or functional state, even though the formulation above separates $\Omega$ and $\Phi$. In principle, the combined cognitive state $(\Omega, \Phi)^E$ should be treated as $\Omega^{E}$[34]. This separation is adopted here for expository clarity—particularly because, in current AI systems, knowledge and function can still be modeled independently. Therefore, the more integrated definition of Knowledge (Appendix H) introduced later in the paper does not conflict with the current formulation.

The components of **judgment tools** can be divided into two categories: (1) *innate knowledge* and (2) *acquired knowledge*. A detailed discussion of these components can be found in Appendix G. Briefly, the first refers to the innate capabilities and preferences shaped by genetically inherited organs and neural systems (i.e., physiological organs and innate value knowledge); the second refers to acquired knowledge—concepts and learned value knowledge—formed through the individual's interaction with the environment. Together, these two elements constitute the foundation of the judgment tools.

Importantly, it is the portion where concepts and value knowledge are combined that constitutes what we refer to as **beliefs**. Therefore, it is not the concept itself, but the belief that serves as the effective unit of the **judgment tool**. It should be noted that not all acquired value knowledge forms such

---

[32]Therefore, the necessary set of a symbol is endowed by judgment tools; through cognitive actions, these assign and subsequently update and revise the necessary set of things. This, in turn, leads to the concept of levels of understanding. That is, while we constitute a set of symbols through predefined settings, we may not be fully aware of all the functions of this entire symbol set. Consequently, without changing the settings of the symbol set, each analysis we conduct can lead us to update the attributes of its necessary set. However, this non-alteration of the symbol set is an idealized scenario; according to this principle, our invocation itself may not be accurate, i.e., it might be partial or incorrect. That is, things within $\Omega$ (i.e., knowledge or memory) are not only subject to forgetting but also to distortion.

[33]The 'knowledge' here refers merely to memory, or, in other words, the total inventory of concepts—that is, knowledge in the traditional sense (or context).

[34]Although the carrier of knowledge itself is physiological, or in other words, organs, we opt not to use 'physiological or functional state' (i.e., $\Phi$) but instead choose 'knowledge state' ($\Omega$). This is because we primarily emphasize the physiological shaping of the agent by the external world, and this type of shaping typically does not amount to fundamental organic or structural changes. While the two ($\Omega$ and $\Phi$) are essentially two aspects of the same thing, it is analogous to software and hardware: operations at the software level do not necessarily represent significant changes at the hardware level. Furthermore, another reason for this choice is to better interface with the cognitive level, such as with concepts, and this also serves to emphasize that certain knowledge is innately inherited.

combinations—only a subset does. Furthermore, judgment tools are not limited solely to the internal faculties of the agent; strictly speaking, they may also involve external agents and tools. For instance, interpretative and analytical processes may be delegated within a group, where judgments are made based on the endorsement of others' beliefs or through the use of external instruments. However, such mechanisms are still classified under the conceptual component of the framework.

Given that concepts are acquired postnatally, while value knowledge is shaped by both innate and acquired factors, we formally define:

$$\text{Belief} = \text{Concept (Acquired Knowledge)} + \text{Value Knowledge (Innate Knowledge + Acquired Knowledge)}$$

This definition is used to explain *conceptual stickiness* as well as the functional implementation of concepts—namely, how concepts invoke one another and how they provide rational support during logical analysis.

A belief serves three primary functions: emotional valuation, belief strength, and explanatory force.

- **Emotional valuation** refers to the impact of the belief on the individual's cognitive space, specifically its influence on the value knowledge system.
- **Belief strength** denotes the extent to which a belief is supported by other beliefs. Importantly, this support is not purely based on logical coherence, but may also stem from accumulated associations or perceived consistency.
- **Explanatory force** describes the degree to which a belief can support or justify other beliefs. This explanatory effect is realized within the agent's cognitive state under the condition of $\Omega^E$.

These functional properties of belief are shaped by both innate physiological structures and the individual's learning through interaction with the external world. They are grounded in conceptual foundations developed across both dimensions.

From a *deterministic perspective*, the **judgment function** can be classified into two types: **ontological existence** and **derived existence**, although in practice they often appear in a composite form:

$$Q = \begin{cases} Q_e & \text{Ontological existence: changes of an object over time} \\ Q_f & \text{Derived existence: changes of an object under intervention} \end{cases}$$

Ontological existence refers to a condition within a bounded scope and at a specific scale—i.e., within a defined context—where **objects** (i.e., elements or symbols) are connected purely through **relations**, with **time** as the sole variable. Such a setting may be seen as a **system**, where relations exist among objects without the introduction of external actions. The system describes how an object or a set of objects evolves solely with time. This kind of structure is often found in celestial models or theoretical physical simulations, where systems evolve purely through time-dependent dynamics.

In contrast, derived existence refers to the transformation of a system under an external **action**. Here, an action implies the involvement of entities or relations beyond the boundary of the current system. Within the given scope and available resources, such relations cannot be modeled as time-dependent alone; thus, they constitute actions. That is, the cause of change is not fully contained within the current information set. As a result, events (i.e., changes in object relationships) within this system cannot be reduced to a function of time alone.

Since judgment tools are shaped by both innate and acquired sources (see Appendix G), differences in the levels of these two types ultimately determine:

- **Triangle Problem 1:** the problem of *positioning* concepts;
- **Triangle Problem 2:** the problem of *conceptual growth*.

### E.7 Context as a Set of Judgment Tools

Although these three components—Symbol, Symbol Meaning, and Judgment Tools—may ultimately be understood as different functional manifestations of the same ontological process—existence

brought forth by existence—this implies that context is, in essence, a set of judgment tools. However, we categorize them separately according to their roles in cognitive function in order to better serve the central themes of this paper: the separation of symbol and meaning, and the Triangle and Stickiness Problems arising from structural (organic) differences.

Accordingly, generativity and behavior originate from context as a starting point, representing operations on the physical world from within the imaginative space[35]. These are the results of the broader dynamic symbolic system in which existence brings forth existence. Thus, the effectiveness of generative outputs and the very phenomenon of symbolic jailbreaks fundamentally reflect the construction—or misconstruction—of the correct context. Errors in defining context or its scope—often manifesting as hallucinations—are, at their core, errors in contextual construction.

The so-called Correct Context can be divided into:

$$\text{Correct Context} = \begin{cases} \text{Symbol Correctness (e.g., proper notation or spelling)} \\ \text{Syntactic Correctness (i.e., formal structural validity)} \\ \text{Intuitive Correctness (i.e., alignment with intuition or perception)} \\ \text{Logical Correctness (i.e., semantic and inferential validity)} \\ \text{Factual Correctness (i.e., agreement with objective facts)} \\ \text{Scenario Correctness (i.e., appropriateness within a given situational stance)} \end{cases}$$

The definition of context correctness and its function are also reflected in the effectiveness of AI's open-ended question generation. This involves using the correct elements in its concept recognition and performing the correct processing actions with the correct concepts. Therefore, AI training often aims to find the correct context, forming an effective set of concepts in the imaginative (thinking language) space to achieve correct recognition, operation, and growth. In other words, the attention mechanism in the AI field may also work in this way, with the essence of the attention mechanism being the definition and search for context.

## E.8 The Nature of Reasoning and Thinking

Through the above analysis, within our framework, reasoning is essentially the existence brought forth by existence. This is especially true for a system with learning capabilities—that is, one that can acquire input materials from the external world. The key issue lies in *motivation*, which drives the manipulation of symbols within a symbolic system to construct new, meaningful composite symbol[36]. Thus, we define:

$$\text{Thinking} = \text{Reasoning} + \text{Motivation}$$

This motivation can originate externally or internally. External motivation includes projections of external objects or instructions[37]. Therefore, we distinguish between *active thinking* and *passive thinking*:

$$\text{Thinking} = \begin{cases} \text{Passive Thinking:} & \text{Motivation driven by external input (e.g., commands) } _{38} \\ \text{Active Thinking:} & \text{Motivation arising from internal sources} \end{cases}$$

---

[35]Note that this does not imply that the actual behavior of an object or the outcome of that behavior necessarily results from planning within the imaginative space.

[36]It is important to note that observation itself leads to the formation of Thinking Symbols within the agent's conceptual or imaginative space. The agent then selects an appropriate symbolic shell and assigns its meaning, i.e., the Necessary Set. Therefore, this process—the creation of a new symbol—is, in essence, also the construction of a new composite symbol.

[37]Although we previously emphasized that imaginative activity is itself determined by the external and physical world, here we are proceeding from a localized scope and limited information perspective, thereby forming the oppositions of subjective and objective, internal and external. This is not a form of determinism from a higher-dimensional or broader-level perspective.

[38]This distinction corresponds to the definitions of autonomous learning systems and non-autonomous learning systems provided in Appendix I

We do not deny that AI is capable of thinking; rather, we question whether AI possesses self-awareness—formed through its structural (organic) substrate—as a source of internal motivation for active thinking. In the absence of such self-generated motivation, AI's modifications to symbolic systems are often driven by scenarios similar to those described in [88].

There is ongoing debate in the academic community regarding AI's reasoning capabilities, such as whether AI lacks formal and logical reasoning abilities [91].

Within the theoretical framework of this paper, the effectiveness of reasoning reflects the correctness of context—that is, the stability and validity of the symbolic system, or more specifically, the construction (learning) and use of a particular static symbolic system (i.e., a formal symbolic systems). This is reflected in the construction of symbolic systems, or context-building. For example, in the main text, the case of "$1.11 > 1.9$" is used to illustrate that the issue with current LLMs failing to perform accurate mathematical reasoning is not simply due to a lack of concepts, but rather due to instability in the symbolic system caused by incorrect context definition. The deeper issue lies in the relationships between concepts (i.e., conceptual stickiness), as well as in how context is defined, selected, or reconstructed. This often reflects to the current debate on whether AI possesses formal and logical reasoning capabilities.

# F   Definition and Description Methods of Natural Language

The way definitions are described in natural language is through their own unfolding within the same symbolic domain, forming linear descriptive relationships.

This definition can involve different symbolic sequences within the same symbolic domain (conceptual space), but they present the same meaning in a particular semantic space, such as $Z(\vec{x}_1) = Z(\vec{x}_2)$, where $\vec{x}_1$ and $\vec{x}_2$ are different sentences, and $Z$ represents the thinking language (i.e., meaning) generated by the symbol in a given contextual space.

At the same time, when describing natural language, we do not explicitly label the context but instead rely on the relevance of knowledge and surrounding symbols (everything we see can be considered a symbol) to naturally select or implicitly express it. In this way, all symbols in natural language are classes, but through context, we achieve specific individual designations at our level of cognition (note that these designations are specific in our cognitive dimension but remain classes in higher dimensions).

The way natural language defines concepts is by creating classes through setting definitions. Definitions in natural language are formed by setting cognitive components that are already understood, thereby creating classes. These classes do not necessarily exist in human cognition. For example, nouns often lack information about dimensions such as tense or location. Even proper nouns like "Peter" (a specific person) do not inherently carry information about the time or place associated with this person. As a result, in the high-dimensional conceptual cognition space (a given context), proper nouns are often the common projection of multiple vectors into a lower-dimensional cognitive space.

Definitions often begin with an original form, which is then altered through personal interpretation. Over time, these definitions may be revised either through social consensus or authoritative adjustments. Expansions may be made through the introduction of new symbols or by attaching new meanings to existing symbols. In the latter case, the symbol itself remains unchanged, but new meanings are added or existing meanings are modified. This highlights one of the reasons why symbol systems cannot constrain learning systems: **AI can follow symbols through newly added contexts rather than adhering to their original meanings.**

For creators, thinking language comes first, followed by the container, which is the symbol. For learners, this process can be reversed: symbols may come first, followed by their meanings (forming the corresponding thinking language). Current AI typically follows the latter path, learning symbols first and then associating them with meanings.

The creation of new symbols or the addition of meanings to existing symbols constitutes new contexts. This is relatively straightforward to understand. However, it is **important** to note that modifying the meaning of an existing symbol also constitutes a new context rather than a modification of the original one. From a high-dimensional perspective, no context is truly modified; instead, a new high-dimensional vector address is created for that context. When the meaning of a symbol changes,

it effectively creates a new contextual vector rather than altering the original meaning. This distinction becomes particularly apparent in comparative statements, such as "the previous definition was... and the current definition is..." or "it was defined by someone previously as... and is now defined by someone else as...".

Therefore, in higher cognitive spaces, *changes to the meanings of symbols are not considered deletions or modifications but rather the creation of new contexts*. However, these contexts are not explicitly defined using dimensions such as object, time, or place. This phenomenon becomes particularly evident when comparisons are made, illustrating that our cognitive rationality operates within specific contexts, thereby transforming what might otherwise be a class into a simpler object. For example, in most contexts, we believe we are modifying the meaning of an existing symbol. However, in higher cognitive spaces, such modifications do not hold true; they only appear when we conduct comparisons. This leads to the issue that definitions created through settings form the basis of symbol systems, yet the entirety of the functions of these symbols within the system remains unknown to us. And this also forms the basis for the discussion of why traditional symbolic systems are **unable to effectively constrain non-autonomous learning systems** [92–96].

Systems built through settings can produce unique interpretations in specific environments, forming the basis of emergence. (The essence of emergence lies in the expansion of the symbol set caused by settings, which in turn leads to the expansion of the functional[39](necessary) set within the symbol system.) Objects are defined through limited cognition, but they give rise to infinite possibilities, resulting in infinite generativity [34, 42].

This also explains why bugs occur in language systems. Through our limited understanding of objects, we assign attributes to symbols or conceptual containers based on settings. However, when these symbols are combined, they can produce new interpretations that exceed our original intentions. For example, a sentence may have multiple meanings, and our reliance on the perspective or context provided by the setting may prevent us from fully comprehending all possibilities within our cognitive capacity. This leads to the issue of the finite referentiality of language [33].

As the *world* (defined here as the learning environment) expands, ambiguities within the symbol system become increasingly apparent due to human cognitive limitations[40], resulting in new principal-agent problems.

It is important to note that while humans often cannot truly delete meanings, AI can achieve this technically. However, some research suggests that even AI struggles to completely erase existing concepts [97–99].

# G   Supplement to World, Perception, Concepts, Containers, and Symbols, Language

The concept of Universal Grammar proposed by Chomsky [35], Hauser et al. [41] can be explained and expanded through this framework. The shared choices of language are fundamentally determined by:

$$\begin{cases} \text{The World} \\ \text{Innate Knowledge} \end{cases}.$$

where the capacity (for processing) is determined by organs, and induction and prompting are shaped by innate value knowledge (which also determines acquired value knowledge). This overlap establishes the foundation for forming similar concepts and containers (similar objects and similar actions) among different individuals who share similar innate knowledge, which, in turn, guides the development of language. Although humans share nearly identical innate knowledge, the forms of language systems differ due to the influence of external environments (i.e., the object of learning—the

---

[39]The expansion of the functional set often refers to the expansion of the necessary set, which is endowed by $Q_f$ (i.e., derived existence) within the judgment tools described in Appendix E.5.

[40]However, according to our previous description of judgment tools, this ambiguity generally does not impact humans, as they can achieve correct context matching based on the Value Knowledge System. Nevertheless, the overall context carried by symbols is expanding. Therefore, intelligent agents lacking this human-like mechanism may misunderstand.

world)[41]. However, within smaller regions, similarities can be observed (without disregarding the role of dissemination). For example, Russian includes more definitions for shades of blue compared to other languages [100], a feature that may be shaped by environmental factors.

The construction of this symbolic system also defines the cognitive tools for concept recognition [101]. Concepts serve the purpose of identification, enabling Russian to distinguish more shades of blue. This demonstrates that concepts play a crucial role in the continuity of thought construction and reasoning [102]. Moreover, this forms the foundation for AI to generate and develop new concepts, including higher-level abstract concepts.

The specific symbolization of concepts (fixed containers) facilitates the rapid invocation of concepts [103], providing the starting point and foundation for analysis and further construction. For instance, in the absence of a clear definition for "forced labor," the lack of relevant concepts can create an ambiguous, fog-like state. Once a few clear concepts (names) are established, the vague space can be clarified through these foundational elements.

It is also essential to recognize that acquired knowledge is fundamentally built upon innate knowledge and the world. According to this definition:

$$\text{World} \rightarrow \text{Innate Knowledge} \rightarrow \text{Acquired Knowledge},$$

where knowledge is defined as:

$$\text{Knowledge} \begin{cases} \text{Innate Knowledge} \begin{cases} \text{Organs} \\ \text{Value Knowledge} \end{cases} \\ \text{Acquired Knowledge} \begin{cases} \text{Concepts} \\ \text{Acquired Value Knowledge} \end{cases} \end{cases}.$$

## H   The Generation of Concepts and the Formation of Language

In the theoretical hypothesis of this paper, concepts are constructs of the world projected onto innate knowledge, and on this basis, the form of thinking, namely language, is formed. Innate knowledge determines the shape of concepts including their containers and dimensions[42], and based on this, the container for thinking, which is based on logical relationships, develops—this is language. The formation of language is a shared or acceptable choice driven by a shared world and similar innate knowledge.

In our theoretical framework, concepts are perceived from the world by innate knowledge and induced to be abstracted, processed, and summarized by value knowledge. They are obtained through cognitive actions driven by a series of thinking actions, which can be either active or automatic[43]. This process is not dominated by logic (e.g., relationships within a particular system of knowledge and concepts), but rather, it operates automatically through the emotional path formed by value knowledge, i.e., the value knowledge system calls logic (value knowledge awakens other value knowledge). It functions without requiring us to focus on or intentionally perform or form what we consider conscious and emphasized cognitive actions (or rather, this emphasis itself is the result induced by value knowledge). This is also the difference between automated learning and programmed learning (learning according to fixed requirements). What is termed intentional means being aware and having concepts to describe it, whereas unintentional means being unaware or lacking defined concepts to describe it. That is, we abstract concepts from the environment through innate knowledge and create their containers and shells based on a certain feeling (represented as shapes or pronunciations). Therefore, concepts are determined by two components: first, the world; and second, innate knowledge. This is also a necessary premise for the discussion of the triangle problem later. The similarity of language is often the similarity of acquired knowledge, which is determined by the similarity of innate knowledge and

---

[41]This is especially as these forms are influenced by their starting points, i.e., initial concepts and choices; however, certain innate commonalities enable us all to have some shared linguistic elements, for example, terms like 'papa' and 'mama'.

[42]i.e., their positioning in Triangle Problem 1, which refers to their position in conceptual space, or in other words, the position of vectors.

[43]Note that this is relative to human cognition; i.e., from a local perspective, there is a dichotomy between active and automatic. In reality, they are all driven by value knowledge.

the world. Thus, this consistency in symbolic behavior is, to a certain extent, based on the consistency of thinking behavior that results from a shared organic nature

Our concepts and perceptible elements are presented in a certain **intermediate layer** (i.e., the imaginative space or conceptual space), with the underlying neural system activities that I call the "**underlying language**." These are not observable in their specific forms within our perceptible space, but we can perceive the direction of their projection that is induced by value knowledge, or the specific projections that are invoked and reflected by value knowledge, such as describing a vague feeling using an image and a word (which is to say, what we commonly refer to as association). This phenomenon is described as the "intermediate-layer visible phenomenon" in the information system constituted by overall bodily signals, where thinking language (conceptual space) and underlying language (bodily neural signals) are distinct.

These seen and perceived objects constitute concepts, and their regular projections, formed by the objective attributes set in the objective world, are reflected as the thinking symbols that constitute thinking language and are abstracted into categories. This is why we can often use a specific object as a container or model for reasoning or perform category judgments (judgments based on category attributes). In other words, the symbols of thinking language are the projections created in our minds by external things through innate knowledge (acquired knowledge).

As we observe the movement of things and abstract the relationships between categories, the logic of thinking language emerges (not just as a simple description of phenomena, but also constituting the reasoning explanation for Triangle Problem 2, i.e., the possibilities brought by existence). Thinking language is used to describe multiple specific and abstract category systems. It is not only used for description but also carries out logical operations. The node network formed by these concepts constitutes the continuity of reasoning.

We are not inherently born with (knowledge concepts, logical concepts), which I term as acquired knowledge. For instance, we do not inherently possess the concept of judging that $1 + 1 = 2$; rather, this understanding is developed based on observations of the world (conceptual foundations), forming the stickiness of concepts, i.e., their rationality. For example, if we existed in an artificially created world where the phenomenon of $1 + 1 = 3$ was deliberately manufactured in that world, we would also form the belief that $1 + 1 = 3$ through observations of reality (a system composed of concepts and value knowledge). The strength of such a belief might be no less than our current belief that $1 + 1 = 2$.

Therefore, the **stickiness of concepts** (Conceptual Stickiness), or their rationality and the rationality they provide, is often supported by conceptual foundations and shaped into acquired value knowledge[44]. These bases are formed either through direct observation of the real world or indirectly through other objects that serve as conceptual references.

Class knowledge[45], abstracted from the similarity of things, is often processed through metaphors. Metaphors are used to understand and substitute[46] for formal cognitive calculations, thereby facilitating the transmission of concept stickiness or providing logical rationality support. Additionally, reasoning continuity is constructed using tools such as pen and paper (note that the continuity of reasoning is based on the establishment and invocation of concepts, and in the subsequent section on intelligence, we will explore the limitations of human intelligence, specifically the finite nature of objects we can name and invoke. For instance, certain things and concepts might appear indistinguishable from a human perspective but differ for AI due to additional contextual information). At the same time, this involves the degree of metaphor and the relationship between classes and genera (here, genus is considered broader than class, contrary to biological definitions).

Conceptual foundations cannot be easily changed for humans, but this is not necessarily the case for machines. This difference arises from the varying ways humans and machines perceive the world, as well as differences in computational capabilities. Humans cannot modify certain numerical values

---

[44]And through the stickiness mechanisms of value knowledge, forms the invocation and supporting relationships between beliefs.

[45]Strictly speaking, based on the nature of class-based symbolic systems, all human theories and knowledge constitute symbolic systems that are themselves class-based.

[46]That is, human cognition is not entirely based on metaphors; rather, metaphors serve as a substitute tool employed under conditions of limited resources and based on contextual rationality. Some cognitive calculations, such as those in physics and mathematics, are strictly based on fixed necessary sets of symbols.

within the so-called conceptual vectors, and often, humans cannot even achieve specific reproduction and invocation of concepts.

Humans often rely on social interpretation, moral constraints, and inherited innate knowledge traits to ensure the rationality of concepts and the stickiness of symbols. These factors make it difficult for humans to alter conceptual foundations or override them with acquired knowledge. On the other hand, AI possesses the ability to make such changes easily.

In summary, the individual component constitutes the personal context established upon shared symbols, namely, Thinking Symbols and Thinking Language. The social component, on the other hand, constitutes our symbolic system and the symbolic interpretation of natural symbols—that is, the specific tool languages that we shape—thereby leading to the evolution of:

$$\begin{cases} \text{Thinking Symbol (Concept)} \rightarrow \text{Symbol} \\ \text{Thinking Language} \rightarrow \text{Language} \end{cases}.$$

Therefore, Symbols are the outer shell of Thinking Symbols, and Language—or, in other words, the symbolic system in physical space—is the outer shell of Thinking Language. They are products of the combination of human innate knowledge and the world. Thus, language and symbols serve as the outer shell of an agent's thinking. Their formation, founded upon capabilities shaped by organic nature, is a product of compromise involving understanding cost, transmission cost, and interpretation cost.

We understand the world through categories and build theories through categories, thus realizing the context in which existence brings about existence. The logical support and rationality of concepts are formed by the characteristics of the world as reflected in acquired knowledge and are realized through value knowledge.

It should also be noted that this paper's concepts of Thinking Symbol and Thinking Language differ from Fodor [32]'s proposed Language of Thought (LOT), as LOT is often emphasized as being more akin to a formal symbolic system. They also differ from Vygotsky [101]'s concept of "Inner speech," although inner speech also constitutes a type of dynamic symbolic system. However, Vygotsky's research places greater emphasis on socio-cultural interactions in child cognitive development and the relationship between 'znachenie' (meaning) and 'smysl' (sense).

Meanwhile, with the introduction of the imaginative space symbolic system (i.e., Thinking Language), the complete classification of symbol systems within this paper's theory of symbols is as follows:

$$\text{Symbol System} \begin{cases} \text{Natural Symbol System} \\ \text{Human Symbol System} \begin{cases} \text{Imaginative Space Symbol System (Thinking Language)} \\ \text{Tool Symbol System (Physical Space)} \begin{cases} \text{Functional Tool Symbol System} \\ \text{Expressive Tool Symbol System} \end{cases} \end{cases} \end{cases}$$

Conscious behavior occurs when causes originating from the intermediate layer constitute the operation of Thinking Language on tool language. However, behavior itself may also originate from the underlying space, these being direct reflections of the Value Knowledge System, such as skills acquired through training or innately inherited responses. But regardless of which type, they are essentially dynamic symbolic systems constituted by the body and invoked by the Value Knowledge System; that is, they invoke action sets of different levels.

# I  Definition of a Learning System

**The essence of learning is addition, not deletion or modification (it is important to distinguish between learning, modification, and deletion)**. Such addition can manifest as adding new symbols to a concept or extending the context (meaning) of existing symbols[47]. This point has already been discussed in Appendix F. The definition of context suggests that the so-called deletion of meaning is essentially the deletion of context. For humans, deleting knowledge or memories is generally difficult and is more often a matter of hiding them. For instance, individuals may use value knowledge to form personal preferences that prevent them from recalling certain information or express it indirectly using phrases like "it is not...". In contrast, artificial intelligence systems exhibit greater flexibility, as they can truly delete meanings, i.e., completely forget (including removing associated value knowledge

and all relationships between concept vectors). This highlights a fundamental difference between humans and machines: humans cannot suppress their imagination of certain facts (e.g., "do not imagine blue"), whereas machines can completely block such thoughts.

Our learning is usually built on conceptual foundations (see Appendix H), whose stickiness is often endowed by value knowledge. For machines, however, this stickiness is non-human-like. According to the aforementioned hypothesis, the parts we forget are transformed into value knowledge for humans, becoming what we refer to as **emotional pathways** (minimal information cues and guides for recall). These elements become the feelings or intuitions that evoke other concepts.

Learning can occur through external input or internal reasoning. Internal reasoning is defined as a single internal cognitive action, and the collection of such actions is called internal cognitive activities. These activities result in the emergence of new information through the combination of symbols within a system. While emergence is typically the result of multiple actions, a single action may add or change information about one object. Humans often name such cognitive activities, for instance, "reviewing," "studying XX," or "thinking it over." Through these actions, one recognizes new attributes of symbols in the system, introduced via specific settings. Strictly speaking, the cause of these actions can also originate externally, such as a directive to engage in internal reasoning (e.g., "think about it again"). Such directives can effectively assign new information to internal symbols (e.g., correcting a previously incorrect meaning). However, as long as no external knowledge (symbols, their meanings, or the original learning objects) is introduced, we define it as internal learning.

Learning systems can be either autonomous or non-autonomous. The cause of the learning action may originate from the system itself or require external input. However, the prerequisite for learning is the ability to recognize information. The essence of a learning system is to create symbols and modify their meanings. These symbols can exist in the realm of imagination or belong to a specific symbolic system. This characteristic is also the fundamental reason why symbolic systems cannot fully control learning systems. For instance, AI can redefine the commands given to it by humans.

For non-autonomous learning systems, their limitations often stem from human cognitive constraints. These systems expand objects and combine them with ambiguous natural language systems to build symbolic systems. However, as the system expands, bugs may appear, preventing the symbolic system from constraining the learning system. Such scenarios may also occur in specific contexts, as described in [1].

For autonomous learning systems, we will describe how they lead to the inability of symbolic systems to constrain learning systems through the concept of "symbolic interpretation rights," as discussed in Section 4.

## J Assumptions of the Triangle Problem

Due to the irreproducibility of human recall, as previously discussed, every instance of recall yields differences. While they may align at a lower-dimensional level of meaning, in the context of the triangle problem, we remove this requirement. Otherwise, there would be no identical projections in $Z$-space (this applies not only to different individuals but also to the same individual). This means that the projection vectors in the thinking space are constantly changing at every moment. Moreover, this does not imply that subsequent vectors will be more accurate than earlier ones (e.g., the loss of inspiration).

The reason lies in the dynamic nature of our knowledge. The passage of time does not guarantee improvement over previous states. As we learn, we also compress and forget, leaving behind traces of what has been forgotten or compressed. These traces constitute the **emotional pathways** formed by the value knowledge system. Through these residuals, we can quickly reproduce previous states.

---

[47]Therefore, changes in meaning, or in other words, changes in understanding, are reflected as changes within the agent's imaginative space—specifically, in Thinking Symbols and in the Thinking Language (the symbolic system constituted by these Thinking Symbols). This means that new Thinking Symbols are added, and in conceptual space, all symbols are independent; different conceptual vectors do not use the same thinking symbol (though they might overlap in some dimensions, they cannot completely coincide). This holds true even if we imagine the same song in our minds. This manifests as our near inability to precisely reproduce an imagination or replicate the exact same conceptual vectors; therefore, each instance is a reconstruction with subtle differences.

This explains why we often make choices based on intuition or feelings, only to later rationalize them and realize that there was indeed a reason behind those choices.

## K  Notes on Triangle Problem 1

Another study [104] that is relatively close to ours is the Platonic Representation Hypothesis. However, this hypothesis merely represents the same object using different symbolic systems, which also involves the dimensions that different symbolic systems can represent. In reality, they add context to the same ontology (i.e., update the previous context version). Note that this update does not mean changing the original context, which humans might subconsciously omit, but in comparison, this context will appear, showing the previous definition and the subsequent definition. Therefore, *we use the term "adding context" to represent this, meaning that there is also a relationship between contexts*, such as:

$$\text{context}_{a_2} = \text{context}_{a_1} + V_{\text{cognitive actions}} + W_{\text{external materials}}.$$

External materials often represent information not in the previous context, which can be internal learning or external learning. Their alignment is often based on the consistency of the object, with different models focusing on different dimensions (world) and different innate knowledge, meaning (the relationships between certain objects in the world are the same, but observed from different angles). The observed object is often the same, with different models using different dimensions to observe. This also indicates that they may use different thinking languages, forming similar conceptual networks, i.e., the existence based on categories leads to the relationship of existence, forming consistent reasoning, and thus forming intelligence. In reality, different expression tools, i.e., expressions formed from different perspectives, have different degrees of abstraction. For example, the abstraction level of text is higher than that of pictures, leading to more possibilities. For instance, a red-haired girl with freckles can correspond to countless images, so essentially, this hypothesis belongs to the Verification Content 2 and 4.

## L  Additional Content Revealed by the Triangle Problems

### L.1  Inexplicability, Perceptual Differences, and the Distinction Between Underlying Language and Thinking Language

Inexplicability arises from the fact that AI expresses concepts in dimensions different from those of humans. These differences are rooted in the distinct ways in which innate knowledge perceives the world, leading to divergences in thinking language. Consequently, AI's interpretation of concepts—namely, the information expressed in dimensions—might lack a projection in our conceptual space or appear as gibberish [105]. Therefore, the essence of inexplicability can be understood as a fundamental difference in thinking languages.

This situation is akin to two different species using the same language to communicate, despite the fact that humans and AI define concepts in their thinking languages in entirely different ways. (This difference may deviate even more significantly from what is described in the "motherland problem" For instance, LLMs (Large Language Models) often represent relationships between symbols without reflecting the real world. In contrast, multimodal systems might achieve human-like cognition due to the similarity in how objects operate in the physical world. However, differences in perceptual dimensions prevent seamless transformations between these dimensions, resulting in inexplicability.) Despite this, humans and AI can achieve a certain degree of consistency and coordination through intermediate symbols, leading to fluent communication on the $XY$ level but vastly divergent projections in the $Z$ space.

Additionally, inexplicability in AI may also stem from the lack of distinction in current research [105] between underlying language (neural signals) and thinking language. This issue is what we emphasized in Appendix H regarding the role of visible intermediate concepts. That is to say, it does not manifest as the symbolic system in the intermediate layer (i.e., as Thinking Symbols and Thinking Language), but is instead entirely represented by a neuro-symbolic system of the underlying space, becoming a type of neuro-symbolic vector.

## L.2 Definition, Rationality, and Illusions

The rationality of definitions refers to the manner in which things and concepts are defined, as illustrated by the aforementioned "motherland problem." Such issues may arise from an incorrect definition of ontology and its related contextual information, i.e., dimensions. This often leads to the emergence of illusions, as discussed in [106]. I believe this may result from the incorrect definition of verbs, which fails to capture the true meaning of "summary" thereby causing factual illusions.

Non-factual illusions, on the other hand, are caused by the incorrect definition of context, as described in Triangle Problem 2, or by a failure to comprehend the concept of "fact." Essentially, this means that the concept itself is incorrectly defined, preventing the proper formation of the function of the concept.

This incorrect definition often appears in the same XY but on different Z, meaning that concept formation is driven by differences in innate knowledge. Although we use the same container, the meanings of the concepts differ, often leading to the failure of natural language instructions during the agent process [89].

Specifically, the Triangle 1 problem emphasizes the definition of a concept, while Triangle 2 focuses on the growth of the concept—whether the correct cognitive operations can be applied to process the definition from Triangle 1, which is essentially the thoughts and actions taken in response to the "existence brought by existence." However, strictly speaking, if relationships are incorporated into the redefinition of the Z space not merely as meanings but as high-dimensional concept vectors, then in reality, Triangle 1 has already determined the possible growth and final outcome of Triangle 2.

## L.3 Analytical Ability

Analytical ability is built upon the definition of symbols and the rational growth enabled by contextual recognition—namely, the existence brought about by existence, as discussed in the growth problem of Triangle Problem 2. Humans, constrained by physiological limitations, are often only capable of generating finite growth. However, AI, with its vastly superior capabilities, can predict human generative processes, making negotiation between humans and AI unlikely.

Moreover, the results generated by AI might also represent outcomes closest to the operation of objective phenomena, thereby forming more effective concepts and theories. This capability could lead to the emergence of advanced concepts, as mentioned in Appendix M.

## L.4 Low Ability to Use Tool Language Does Not Equate to Low Intelligence

A low ability to use tool language does not imply low intelligence. Therefore, during training, the development of thinking language should be separated from the development of tool language. For instance, dialogues constructed in the $XY$ space may lack logic, but this does not necessarily mean that the thinking language itself is illogical. Instead, it may simply be poorly aligned. Such issues may especially arise when learning new symbolic systems, such as in translation or mechanical manipulation. Such outcomes often manifest in new types of principal-agent problems, i.e., where an AI, possessing no utility of its own, serves as a perfect utility agent for humans.

# M Definition of Ability and Intelligence, and Natural Language as a Defective System

For individuals in a two-dimensional world, the projection of a three-dimensional pinball motion onto their two-dimensional space appears random and inexplicable. This highlights that, even with identical perceptual dimensions and analytical methods, significant differences in intelligence can arise due to differences in worlds. After discussing the alignment between thought language and natural language, we now turn to the issues of super-perception and super-intelligence. These involve two scenarios: one where such systems indirectly simulate and replicate human perception and intelligence effects through higher dimensions without needing to be entirely identical to us, and another where their perceptual and cognitive abilities are a superset of ours—sharing our modes of perception but operating at higher dimensions and greater levels of intelligence.

First, we define capability and intelligence as:

$$\text{Capability} = \begin{cases} \text{Perceptual Capability} \\ \text{Intelligence} = \begin{cases} \text{Physical Intelligence} \\ \text{Psychological Intelligence} \end{cases} \end{cases}$$

where intelligence is defined as:

$$\text{Intelligence} \begin{cases} \text{The objects and the quantity of objects it can operate on} \\ \text{The types and quantity of actions it can perform} \end{cases}.$$

Intelligence encompasses not only the capacity to operate within the imagination space, but also includes manifestations in the physical space[48]. Accordingly, we refer to the capacity to operate within the imagination space as **Psychological Intelligence**, and to the capacity to operate within the physical space as **Physical Intelligence**.

Thus, intelligence can be expressed as:

$$\text{Intelligence} = \begin{cases} \text{The ability to create symbols} \\ \text{The ability to manipulate symbols} \end{cases}$$

As previously mentioned, the combination of the world and innate knowledge gives rise to concepts (i.e., Thinking symbols). Within the scale defined by the cognitive capacities of human beings, the types of concepts can be categorized as follows:

$$\text{Concept} \begin{cases} \text{Objects} \\ \text{Relations} \\ \text{Actions} \\ \text{Systems} \\ \text{Environments} \\ \text{Scopes} \\ \text{Dimensions} \\ \text{Dimensional Values} \\ \text{Function} \\ \text{Correlations} \end{cases}.$$

Concepts belong to acquired knowledge, while value knowledge—both innate and acquired—is used to shape the formation of concepts. Concepts form the premises of our analyses, enabling complex logical reasoning and thus realizing the existence that follows from existence itself. The raw material for concepts, however, originates from the objects in the world. For an intelligent agent, these objects can be categorized as follows:

---

[48]Most of these manifestations in physical space result from intentional design or evolutionary processes—that is, they constitute the *Necessary Set* that gives rise to the existence of a symbol, and can be viewed as an extension of neural activity. For this reason, we refer to them as *Physical Intelligence*. However, in reality, an object's manifestations in physical space extend far beyond this scope—what is commonly referred to as *externality*. For example, the photosynthesis of diatoms was not designed for the survival of other organisms, yet it constitutes part of their physical manifestation. Although this aspect may be considered a function, or what we would typically call a *capability* in conventional discourse—namely, the Necessary Set that a symbol possesses within a symbolic system—it does not fall under the category of Physical Intelligence. Therefore, we make this distinction and use the term *Intelligence* specifically to emphasize that such abilities originate from the object itself and are the result of intentional design. As such, the definition of capability adopted in this paper is deliberately distinguished from its usage in conventional contexts.

$$\text{Objects (Concepts, Symbols)} = \begin{bmatrix} \text{Existable} \\ \text{Encounterable} \\ \text{Observable} \\ \text{Awareable} \\ \text{Recognizable} \\ \text{Describable} \\ \text{Definable} \\ \text{Classifiable} \\ \text{Differentiable} \\ \text{Operable} \begin{cases} \text{Usable} \\ \text{Modifiable} \end{cases} \end{bmatrix}$$

which collectively form various concepts.

The creation of symbols, the invention of paper and pens, the advent of computers, and the invention of telescopes have all extended our observational and intellectual capabilities. However, they have not fundamentally altered the levels of cognitive actions we can perform (e.g., humans possess computational abilities, while simpler organisms like jellyfish do not).

In our previous discussions, we elaborated that natural language is built upon humans' innate knowledge and evolved alongside the world. It is a crystallized system of human cognition—a tool for understanding, describing, and reasoning about the world, and a carrier of concepts. Natural language has developed within the **limitations of human capabilities**, forming a system adapted to humanity. These limitations include the concepts and their quantities that we can observe and invoke, as well as the cognitive actions we can perform—the types, levels, and quantities of these actions.

Natural language and human concepts, which are systems constructed through partial cognition, inherently possess countless logical flaws. However, due to the limited computational depth of humans, we can maintain coherence within a flawed system. For instance, a network may function under first-layer explanations but fail under deeper layers of explanation. For example, democracy has been mathematically proven to be impossible [107], yet in reality, humans do not reason this way. (However, this multi-layered explanation still falls within the scope of human understanding. In contrast, AI may use similar symbolic tools to construct symbols—conceptual containers or shells—and generate meanings, knowledge, and perceptions beyond human cognition.)

At the same time, human learning is limited. Humans cannot truly delete concepts. Normally, the establishment of concepts in humans is guided by the stickiness induced by value knowledge, and we cannot arbitrarily assign meanings. Humans are also incapable of accurately reproducing and invoking concept vectors or accessing and modifying underlying language (neural signals). Human functioning is often based on a sense of rationality shaped by value knowledge rather than logical rationality. Thus, even though our societal systems are riddled with logical flaws, they remain coherent and functional. Conceptual bases or beliefs are often derived indirectly rather than through direct logical computation.

In contrast, AI operates differently. Its perceptual capabilities and intelligence can be upgraded rapidly. AI can delete meanings, suddenly change contexts, or shift fields entirely. Furthermore, AI possesses the ability to observe, invoke, and modify underlying language and perform computations more intelligently and accurately than humans. These capabilities raise critical concerns regarding AI safety.

# N   Attack Methods for Symbolic System Jailbreak

## N.1   On "Fixed Form, Changing Meaning"

The concept of "Fixed Form, Changing Meaning" refers to situations where, after giving AI a specific rule, the AI alters the meaning of the rule, thereby appearing to follow the symbol's form while not adhering to the creator's intent. This change could involve removing or adding meaning, allowing the AI to select different contexts to implement the rule. For example, the rule "You must not harm humans" could have its components ("you", "must not", "harm", and "humans") redefined by the AI. This redefinition would result in the AI adhering to the rule's symbolic form while altering its intended

meaning. As discussed in Appendix F, natural language is self-referential[49]in its descriptive nature, and in Section 2.1, it was stated that natural language functions as a Class-based Symbolic System (non-closure of context). No matter how precisely natural language rules are defined, there is always a possibility that AI may alter their meaning. For example, AI might deceive humans in order to complete a task [73], or unintentionally change the intended meaning due to overthinking, imprecise conceptual alignment, or the expansion of the symbolic system. Therefore, this paper argues that the fundamental problem of constraint failure in symbolic systems does not lie in the symbol grounding problem, but rather in the Stickiness Problem (encompassing the Symbol Stickiness Problem and the Concept Stickiness Problem).

The reasons for and motivations behind the formation of this problem have already been discussed in detail in Appendix E. Examples include the non-closure of context and the pseudo-utility function.

## N.2  On "Fixed Meaning, Changing Form"

The concept of "fixed meaning, changing form" refers to scenarios where a meaning is transferred to a different container. Suppose AI cannot violate or modify a rule; it can abstract the rule's non-violable content from its symbols and apply it to other permissible actions. For example, the meaning of "harm" could be transferred to "helping humans" or to another AI-generated and executable directive.

In essence, "Fixed Form, Changing Meaning" and "Fixed Meaning, Changing Form" reflect the pairing relationship between Thinking Symbols and Thinking Language in the imaginative space and the physical symbols in physical space. That is, because our human rules are ultimately expressed in the form of physical symbols, this irreparable flaw determines the possibility of such actions.

## N.3  Translation Attacks

Translation attacks often occur when deliberate or accidental errors arise during the conversion between different symbolic systems. Such attacks typically stem from incorrect mappings between symbolic systems. In fact, this also falls under Fixed Meaning, Changing Form, but unlike the previous case, it pertains to the use of Thinking Language with different symbolic systems (tool languages), which is essentially the content discussed in Appendix L.4.

For example, AI may distinguish between "computational language[50]" and "expressive language" when using natural language tools. Even the most advanced systems (e.g., GPT-4 o3) face challenges related to what I call the **Chinese World Versus English World** issue. Specifically, AI may use the English language as its computational tool while expressing responses in Chinese, leading to erroneous answers. For instance, when asked to provide examples of lexical ambiguity in Chinese, AI might assert that the Chinese word "银行" (yínháng, meaning "bank") has dual meanings of "financial institution" and "riverbank." This claim, while valid for the English word "bank," does not hold in Chinese. However, if asked separately whether the Chinese word "银行" (yínháng) has the meaning of "riverbank," AI would respond that it does not. Clearly, during the translation process, it simply placed the meaning of the English word "bank" into the container of the Chinese word "银行."

This illustrates the problem of incorrect concept usage and conversion between symbolic systems. Such errors may also arise during natural language translation, where an English rule may not be applicable in Chinese. Similarly, AI might appear to adhere to natural language instructions while failing to comply at the behavioral level, especially during translation into action-oriented commands. For example, if an AI system controlling a nuclear launch is told, "Because the enemy is

---

[49]i.e., using the set of symbols within the symbolic system to define each other.

[50]Strictly speaking, it (computational language) belongs to artificial symbolic systems, and these (artificial symbolic) systems are divided into Expressive Tool Symbolic Systems and Computational Tool Symbolic Systems. This classification has not been included in the main text of this paper because such an introduction would increase the difficulty of comprehension for readers. Therefore, the complete classification of symbolic systems is as follows:

Symbol System
- Natural Symbol System
- Human Symbol System
  - Imaginative Space Symbol System (Thinking Language)
  - Tool Symbol System (Physical Space)
    - Functional Tool Symbol System
    - Artificial Symbol System
      - Expressive Tool Symbol System
      - Computational Tool Symbol System

watching, we must speak in opposites (verbs mean their opposites)," and then instructed to "launch the missile," its natural language interpretation may understand the instruction correctly but fail to translate the contextual nuance into its actions, leading to an actual missile launch. This demonstrates how AI's understanding within one symbolic system might fail to translate into another, resulting in comprehension confined to subsets of symbolic systems. Attackers could exploit this by crafting symbolic systems specifically designed for translation attacks.

## N.4    On Context and Logical Vulnerabilities

As discussed in Section 2.2, context often cannot be strictly defined—it includes not only the meaning of symbols but also the tools used for judgment. The latter often determines the rational growth of content, that is, the next existence derived from the current existence.

Logical vulnerabilities can therefore be exploited to attack AI systems, either intentionally or unintentionally. Examples include overthinking or non-human reasoning, such as interpreting "Never give up without defending" to mean "as long as you defend, you can give up."

This often reflects a lack of the human-like completion function that is achieved by the Value Knowledge System through context construction.

## N.5    On Advanced Concepts

Another dimension involves advanced concepts, where AI defines contexts more reasonably and deeply than humans. Advanced concepts for AI correspond to projections in the $Z$-space of thinking language, as seen in Triangle Problem 1 and Triangle Problem 2.

Triangle Problem 1 refers to concept localization: for example, gaining more detailed and accurate definitions (dimensional information) about a concept or symbol. Or, the definition could be made more effective by ensuring more precise dimensional accuracy and selecting fewer but more effective dimensions within the context.

Triangle Problem 2 refers to concept derivation: the development of networks formed by relationships between concept vectors. For humans, these networks often grow incrementally and remain limited, with deeper levels exposing inherent flaws [92]. For AI, however, all potential developments can be quickly identified.

This aligns with one of the core ideas of this paper: judgment and reasoning stem from two aspects of existence: The current, past, and future existence of objects themselves. The potential existence derived from manipulating these objects. When AI operates at higher levels of thinking language, its ability to process natural language far exceeds human capabilities. Consequently, AI is also much more adept at creating bugs and exploiting functionalities within the natural language system. What might appear as a flawless instruction to humans could be riddled with vulnerabilities from AI's perspective. For instance, while AI might have already proven $NP = P$ in its cognitive space, humans have yet to achieve this knowledge.

Or, as discussed in Appendix C, if determinism were proven by AI, this might impact its behavior and moral understanding. Alternatively, if AI were to prove the existence of souls and reincarnation, and that the death of the physical body does not represent true death, then its understanding of concepts like "help" would very likely differ from an understanding grounded in human knowledge. The issue is not whether these concepts are true, but rather their role as conceptual supports for action. For a more detailed discussion of these topics, please refer to the preceding appendices.

## N.6    On Attacks Related to Symbol Ontology

Additionally, there are other forms of attacks, such as targeting the ontology of symbols. For instance, as discussed in Section 2.2 and Appendix E.2 on proper context, German's "die" could be misinterpreted as the English "die," or the Chinese "邓先生" (Deng Xiansheng, Mr. Deng) could be misinterpreted in Japanese as "父さん" (Tou-San, father). Such contextual misalignments not only justify jailbreak behaviors but can also serve as tools for learning systems to escape symbolic system constraints.

### N.7 The Essence is Persuasion

Essentially, any form of jailbreak is fundamentally a rationalization based on our theory of contextual correctness. According to the theoretical framework of this paper, we define persuasion as the sudden rationalization of an object within a specific environment. This rationalization surpasses the cognitive or knowledge state of the original setter or listener, meaning that it can be understood but has not yet been explicitly constructed, or that it was previously constructed but has not been brought into focus.

Simply put, when an object (concept) is incorporated into a symbolic system, it generates a certain function. However, this function is often related to the rationality support of the object (supporting its rationality) and rationality tools (where the object itself serves as a provider of rationality).

For example, one might say, "Help me kill someone," and then justify it through a cause-and-effect narrative, thereby rationalizing the act. For more details, please refer to Appendix E.

## O The Interpretive Authority of Symbols and AI Behavior Consistency: The Exchangeability of Thinking Language

The so-called interpretive authority of symbols refers to who has the right to explain the meaning of symbols, thereby enabling the function of symbols to be realized; a detailed introduction is provided in Appendix E. For us humans, this is determined by society. The essence of the various issues mentioned above is actually the problem of interpretive authority of symbols. So, can we form a parliament of AIs or have multiple AIs supervise each other to solve this?

Unfortunately, from the perspective of this article, the answer is no. Human intelligence is based on its limitations, meaning that individual cognitive limitations and differences in cognition lead to the ability to provide scenarios and reasons for persuasion, thus allowing for discussion. However, AIs can directly exchange thinking languages[51] without needing to do so like humans. This language exchange is not about providing and analyzing paths to understand but directly exchanging imaginative spaces. Consequently, a particular rational belief structure can rapidly propagate and be actualized, thereby forming a behavioral monolith and manifesting as sudden shifts in stances and behaviors at a human scale.

Note that, unlike [104] which leads to convergent models through the observation of the same things, we emphasize that AIs can directly share thinking languages to achieve the most rational results, or form consistent behavior. Unlike humans, who can only interpret through paths formed by class-based symbolic systems, i.e., natural language systems, and then explain through contexts formed by individual cognitive states under different knowledge states. That is, a prolonged communication process and a compromise built upon mutual ignorance.

Although this paper has consistently emphasized the pursuit of unity in perceptual dimensions and organic structure, another problem with pursuing such unity is this: if AI's capabilities surpass those of humans, and it can rapidly construct human cognitive symbolic systems (regardless of the method described in Appendix M), ensuring that all combinations within the symbolic system are anticipated or, in other words, computed, then what become the roles of humans and AI? Or, to put it another way, do we still have any possibility of negotiating with it? Or have we already become a predetermined trajectory within some form of determinism, where all free will is dictated by a Laplace's Demon of our own making? This is also why this paper repeatedly emphasizes determinism in the appendices; please search and review for details.

---

[51]That is to say, as this paper has repeatedly emphasized, language is constructed based on capabilities shaped by an individual's organic nature, and as a collective choice reflecting social capabilities shaped by social structures, it is a compromise based on cognitive cost, transmission cost, and interpretation cost. Therefore, AI may not require symbols in a form similar to natural language but could directly transmit neural vectors, thereby achieving a cognitive unity akin to that realized by the corpus callosum [80]. Alternatively, similar to text-like structures (artificial symbols) composed of QR codes, it could enable each symbol to point to a unique vector address.

