# OpenReview forum: "Rules Created by Symbolic Systems Cannot Constrain a Learning System"
_NeurIPS.cc/2025/Position_Paper_Track — Submitted to NeurIPS 2025 Position Paper Track_

### Official Review · Reviewer_28d2 · 2025-08-12

**Significance:** 2
**Presentation:** 2
**Rating:** 4
**Confidence:** 3

**Summary:**

The paper argues that rules expressed in symbolic form cannot constrain a learning systems, since AI can modify the meanings of the symbols by creating new context.

The problem is not that the symbols are not grounded, but that symbols are not sticky (the binding between the symbol and its meaning can change depending on the context).

The problem is illustrated with the triangle problem.

Finally the paper calls for the establishment of a new field addressing symbol-related risks in AI, ensuring that human intent is aligned with the behavior of AI.

**Strengths:**

This is certainly a interesting paper that takes a birds' view on some fundamental challenges of AI systems.

**Weaknesses:**

While the main message can be understood by a broad audience, the discussion is often heavy.

The authors state the paper theoretically demonstrate the problem, the proof is not formal in any way, philosophical perhaps.

There is no discussion of alternative positions, and there is a lack of real-world evidence.

**Questions:**

Do you think that currently there are no real world AI systems that strictly aligns with the human intent (disregarding sub-optimal optimization)? If there are, what are the conditions for achieving alignment? How can we quantify the misalignment, and how to discern from sub-optimal optimization?

Some of these questions belong more to the suggested new field rather than strictly concerning the content of the paper. However, they do relate to the lack of discussion of alternative positions.

**Alternative Position:**

No

**Author Identification:**

No.

**Context:**

2

**Discussion:**

3

**Ethics:**

["NO or VERY MINOR ethics concerns only"]

**Position:**

Yes, the paper argues for or against a position related to machine learning.

**Support:**

2

**Thoroughness:**

3

---

### Official Review · Reviewer_ZwDU · 2025-08-16

**Significance:** 2
**Presentation:** 1
**Rating:** 3
**Confidence:** 4

**Summary:**

The central argument is that symbols are inherently meaningless, and their meanings are assigned and modified through learning and context. An AI, therefore, can bypass any rule-based constraint by altering the meaning of the symbols used to express that rule. The introduce : "Stickiness problem" - one can hack the meaning of a symbol by virtue of the context its presented; and the Triangle problem - two entities can communicate but this doesn't guarantee that there are no underlying conceptual differences.

Their argument is -> symbols are flawed (because of stickiness & triangle problems) and while it worked ok for humans (I'm not too sure of this argument but please correct my understanding) - AI can misinterpret symbolic meaning (or rather hack the meaning for ulterior gains)  causing AI safety concerns.

**Strengths:**

The paper is provide a good number of examples and references - and detailed discussion on stickiness & triangle problem. The occasional examples are also helpful to understand their point.

I also, generally, agree with the intent of the position i.e. symbols have lots of challenges and we must realize its challenges in constraining AI systems. It's also quite topical given the recent popularity of LLMs ( and claims on reasoning, theory of mind etc. ) so much so that the impacts of LLMs is materializing.

**Weaknesses:**

1. The writing is hard to follow. The paper does contain all the pieces (examples, definitions etc.) but the overall structure is convoluted. It's very hard to point out the motivation of a section and decouple author's novel claims / arguments from existing work. They also inter-twin definitions with making an argument using that definition very often. In general, a clear flow is missing or atleast very hard for to follow.

2. Novelty of the position : Maybe the authors can clarify / refine their position.
(a) They argue that symbols have challenges with grounding. This is already known to the community.
(b) A refinement is - the challenge stems not just from grounding but rather from stickiness. - Can the authors differentiate b/w stickiness & grounding in their rebuttal. Grounding is the inability of precisely define real-world connection through a symbol. This is because conveying entire space-time tube is hard, and even so, conveying mental models will always be lossy. Stickiness - as is binding b/w symbol and meaning. What is the key difference b/w the two terms that authors use to highlight their position?

3. Triangle problem seems to be a refined version of Chinese room argument. If so, this adds to the weak novelty.

**Questions:**

(see weaknesses as well)

4. Reward hacking is a known & popular issue - which happens because AI effectively "games" the specification. A popular reason is, incomplete specification from the perspective of AI (because we didn't provide common-sense, our mental model and space time tube) and AI systems find ways of hacking. What does the symbol stickiness argument add to this known issue.

5. The above points are also raised in the paper in some form (like jail break, path media ..., etc.) While the authors use it as a supporting argument, it adds to writing quality issues (decoupling motivation from a clear novel argument of their own)

6. (Position papers do not have to provide a solution). While the observation that lack of space time tube (& my argument - lack of conveying mental models) is valid, accepting this view doesn't provide any path forward. Regardless of advancements, with all the sensory overload, conveying mental models will always be lossy (authors don't use this argument).

**Alternative Position:**

Yes, and alternative positions are well-considered and named but not addressed

**Author Identification:**

No.

**Context:**

2

**Discussion:**

3

**Ethics:**

["NO or VERY MINOR ethics concerns only"]

**Position:**

Yes, the paper argues for or against a position related to machine learning.

**Support:**

3

**Thoroughness:**

3

---

### Official Review · Reviewer_rmAn · 2025-08-31

**Significance:** 3
**Presentation:** 3
**Rating:** 4
**Confidence:** 3

**Summary:**

This paper claims the assumption that symbolic rules, e.g., laws, constraints, and formal verification can effectively constrain AI. They argue that symbols are inherently meaningless and may be reinterpreted, leading to the Stickiness Problem. They introduces the Triangle Problem to demonstrate the disconnect between symbolic constraint and conceptual equivalence. The main claim is that AI lacks human-like perception and cost mechanisms, thus it will inevitably reinterpret symbols and bypass constraints. They further call for the establishment of a new field “Symbolic Safety Science” which focused on symbol-related risks in AI alignment.

**Strengths:**

1. The paper propose a quite interesting and promising perspective on symbolic limits of AI alignment.

**Weaknesses:**

1. While the paper offers valuable conceptual insights in symbolic language system, concepts and communication, it lacks practical examples and experimental validation in the AI safety especially jailbreak domain, which would be important to emphasize its importance. It would be valuable to design heuristic experiments to validate whether some of these safety concerns arises from difference in conceptual space and reassignment of symbols’ meanings, as suggested in the paper. Some concepts such as Context or Path Media are too broad and too detailed which carries AI area irrelevant contents. As a result, the AI safety section receives less attention.
2. The writing is not clear enough, and many contents are too obscure and difficult to understand. For example, the so-called stickiness problem and triangle problem, as well as their relationship, are written in a rather confusing way.
3. The author's viewpoint is novel, but symbol-related risks remain an conceptual problem, and the author does not seem to provide substantive suggestions for establishing "Symbolic Safety Science" based on the Stickiness Problem and Triangle Problem.

**Questions:**

1.	Could the author explain more about how Symbolic Safety Science concretely differ from alignment and interpretability research?

2.	Can the Stickiness Problem be tested empirically on existing LLM jailbreak phenomena?

3. How can we determine whether some AI safety concerns are caused by different conceptual space or reassignment of symbols? Are there any preliminary guidelines for designing experiments to test this?

4. Are there any opposite claims which argue that symbolic constraints alone are enough for AI regulations? What are their considerations, and why do you find them unreasonable in your reasoning?

5. Is there any research on cognitive difference between humans and LLMs that could support/challenge your claims?

**Alternative Position:**

No

**Author Identification:**

No.

**Context:**

2

**Discussion:**

3

**Ethics:**

["NO or VERY MINOR ethics concerns only"]

**Position:**

Yes, the paper argues for or against a position related to machine learning.

**Support:**

2

**Thoroughness:**

3

---

### Note · Authors · 2025-09-04

**1-11 Submit Again:**

Unsure

**1-1 Submission Process:**

2

**1-2 Next Year:**

1. Establish and disseminate specific reviewer guidelines and regulations for position papers to both authors and reviewers.
2. Provide a clearer and more explicit timeline.
3. Relax the word count limit to allow authors to express their ideas more thoroughly and politely. (We apologize if some of our expressions seemed impolite due to the strict word count constraints.)
4. Implement a mechanism to screen for intellectual conflicts of interest, for example, by allowing the specification of keywords for such conflicts.

**1-3 Future Development:**

1.  We previously sent，from rcbssccals@gmail.com，an email titled “[NeurIPS 2025 Position Track | Submission #430] Cross-Conference Suppression & COI — Statement and Request for Assistance,” detailing the heightened ethical risks currently facing the Position Track due to academic conflicts—for example，our report alleging cross-conference suppressive reviewing by Reviewer ZwDU. We also noted that the absence of a rebuttal stage and a policy under which only accepted papers may be discussed publicly can harm the basic academic rights of junior scholars by obscuring deliberate suppression and appropriation. In particular，when established academic authorities who are direct competitors both submit position papers taking an opposing stance and simultaneously review conflicted submissions by junior scholars—issuing disparaging evaluations—there are inadequate protection channels for those juniors，and the likelihood of collusion between ACs and reviewers increases. This harms the fundamental academic rights of early-career researchers. It is unfair and unethical，and it is inconsistent with the spirit of scholarship and the healthy development of the community.

2. To address reviewer scarcity，we propose appointing junior reviewers drawn from early-career scholars. Their responsibilities would be limited to writing summaries，performing typo/terminology checks，and compiling reports of policy violations observed during review—thereby cultivating a pipeline of junior reviewers.

3. We propose establishing a Young Scholars Position Track to promote peer exchange among early-career researchers，ensuring that position papers are no longer a one-way publication channel and privilege reserved for established authorities.

4. For managing serious conflicts of interest rooted in intellectual conflicts，we recommend following guidance such as article in ACM SIGARCH，“Ethical and Moral Fraying due to Intellectual Conflicts in Paper Reviews,” and adopting concrete procedures accordingly.

**1-4 Interest:**

["Panel discussions with other position paper authors", "Structured debates on controversial topics", "Workshops for developing position papers", "Mentorship programs for early-career researchers", "Other (please specify in the next question)"]

**1-4 Other Interest:**

1. Discussion on academic fairness and the protection of junior scholars’ academic rights

**1-5 Thoughtful:**

3

**1-6 Supportive:**

6

**1-7 Technical Aspects Versus Position:**

2

**1-8 Gate Keeping:**

2

**1-9 Camera Ready Changes:**

1. Clarify scope and positioning. Explicitly state that the focus is black-box AI and learning systems, distinguishing them from interpretable (glass-box) models designed for auditable deployment with offline retraining and re-validation. Make clear that, prior to large-scale deployment, any theory or study that purports to underwrite the safety of black-box AI or Post-hoc XAI must confront and answer these two questions. Also clarify this paper’s status as a position paper—its purpose is to pose these two new questions (see our response to the reviewers).

2. Emphasize implications for glass-box AI. Highlight that the problems raised here—viewed from a new perspective and theoretical basis—support the necessity and superiority of the glass-box AI development path and respond to the debates and technical approaches for which new citations will be added; glass-box AI is naturally immune to these two problems and can better serve human welfare.

3. Restructure and revise. Move Appendix A (Alternative Views) into the main text and—based on the two points above—make light revisions to the Abstract, Introduction, and Conclusion, etc., to specify the discussion target, refine the tone, and incorporate adjustments in light of the reviews.

4. Citations for the Triangle Problem experiments. Add supporting citations in the experimental section on the Triangle Problem.

5. Additional citations. Add citations to the following:

	1. Stop explaining black box machine learning models for high stakes decisions and use interpretable models instead

	2. Interpretable machine learning: Fundamental principles and 10 grand challenges,

	3. Concept whitening for interpretable image recognition

	4. The right to a glass box: Rethinking the use of artificial intelligence in criminal justice

	5. Why black box machine learning should be avoided for high-stakes decisions, in brief

**3-1 Review Response1:**

Reviewer 28d2

**3-2 Reaction To Review1:**

1. This paper examines the new risks created by combining black-box AI with learning capacity，which changes the role of symbols. Unlike traditional formal procedures，constraints are no longer implemented by rules formed from symbols. Accordingly，any theory or study that purports to underwrite black-box AI and learning systems must，before real-world deployment，answer two questions: (i) the Stickiness Problem (how to prevent already-trained symbols from being assigned new meanings)，and (ii) the Triangle Problem (how to ensure human–AI consistency in thinking language so that symbols can function as intended). This preempts a new principal–agent problem and is not a matter for immediate experimentation

2. We point out that governance approaches for black-box AI—whether via training such as RLHF or via external-structure constraints (e.g.，LLM-Modulo)—face novel risks introduced by these two problems (due to page limits，discussion of alternative positions appears in Appendix A). Black-box AI constitutes a form of heterogeneous rationality distinct from humans. Human symbolic systems are the outward expression of our capacities，perception，and thinking language by which we cognize the world. Consequently，attempting to govern black-box AI by issuing symbolic instructions or imposing external structures runs into these two problems. By contrast，statically deployed glass-box AI is naturally immune to both; framing these two problems thus offers a new argument that glass-box AI can better serve human welfare and safety

3. For governance of learning-based black-box AI，we do not adopt methods that correct behavior through symbols，so as to avoid defects of human symbols (meaning separability，polysemy，non closedness). Achieving human–AI model reconciliation through symbols，in fact，manufactures the Triangle Problem. We therefore advocate coordination via BCIs that enable direct readout of human neural vectors，and use this as the starting point for designing capabilities and rules

**3-3 Review Response2:**

Reviewer ZwDU

**3-4 Reaction To Review2:**

1. AI does not "hack the meaning for ulterior gains." As defined by the new principal-agent problem (p9)，even an AI that has no utility of its own and acts merely as a projection of human utility can still harm human utility due to the Stickiness Problem and the Triangle Problem

2. Human symbols and language are formed on the basis of human perception，capacities，and innate preferences; they are the external manifestation of human thought. Although flawed，they function within human systems

3. We never states "grounding is the challenge". The Stickiness Problem concerns what should be retained and what should be rejected during learning; it involves the difficulty and legitimacy of creating new symbols and concepts and of adding/modifying symbol meanings and concepts. Its sources are detailed in Sec. 4 (e.g evolved human sociality，society’s interpretive authority over symbols). Symbol stickiness problem refers to symbol–meaning separation and to one symbol mapping to multiple concept-vector meanings. Therefore，even if a symbol’s meaning has been correctly trained，by virtue of its learning ability an AI can still assign new meanings to an already grounded symbol，thereby changing its meaning

4. Reward hacking refers to an AI’s failure to act according to human intent behind symbolic instructions; it does not involve modifying symbol meanings. It indicates holes in the semantic net woven by symbols，whereas the Stickiness Problem says the net itself can be removed or altered

5. The Triangle Problem refers to differences in the concept-formation process caused by architectural and capability differences; it emphasizes the separation between internal thinking language and external tool language. Even consistency of symbolic behavior cannot mask differences in thinking languages，which leads to deviations in subsequent behavior and outside the training environment. It is entirely unrelated to the Chinese Room argument

6. Your viewpoint is flawed. see split-brain syndrome

**3-5 Review Response3:**

Reviewer rmAn

**3-6 Reaction To Review3:**

1. Positioning：This paper addresses the AI community from the perspectives of other disciplines and communities—perspectives the AI community has lacked or overlooked—leaving some topics under-discussed. We therefore must introduce interdisciplinary discussion，making these issues ones that require careful reading to understand，which also helps explain why they previously went unrecognized and undebated. We set out the problems that widespread real-world deployment of AI will face，call on the community to take them seriously，design experiments around these two problems，and jointly plan a Symbol Safety Science to address risks arising from the changing role of symbols brought about by black-box AI and learning. Accordingly，given limited space，this is a problem-posing position paper，not an experimental paper.

2. Scope：Alignment and interpretability research is a subset of Symbol Safety Science (p. 37).

3. Detection and experiments. For concrete detection and experimental work，one can build on Concept Whitening，ProtoPNet，and SegDiscover to detect Stickiness Problem arising from learning capacity; combined with brain–computer interface (BCI) techniques，and using the four Verification Content items proposed on p. 7，one can test whether the concept dimensions and values of the thinking language are aligned with humans and how they vary across learning and behavior. This substantiates the Triangle Problem (p. 38) and the symbol jailbreak detection on pp. 41–43，providing stronger theoretical support for the safety case for glass-box AI.

4. Governance approaches such as Kambhampati’s Model Reconciliation and LLM-Modulo do not resolve the Stickiness Problem or the Triangle Problem.

5. Theoretical grounding：Starting from Perceptual Symbol Systems (PSS) theory，We examine how AI and humans differ in concept formation; there is already substantial theoretical support for our view，e.g.，“From Tokens to Thoughts: How LLMs and Humans Trade Compression for Meaning.”

6. can't..AI

---

### Meta-Review · Area_Chair_J4tx · 2025-09-08

**Rating:** 3
**Confidence:** 4

**Strengths:**

The reviewers agree with the authors on the importance of the highlighted perspective on symbolic limits of AI.

**Weaknesses:**

All reviewers find the paper's writing and arguments difficult to follow. One reviewer criticizes the proposed Symbolic Safety Science not sufficiently substantiated and comments that the usage of certain concepts (such as context, path media) too broad. Another reviewer mentions the difficulty in following the motivation of a section and distinguishing the author's novel claims  from existing work. Furthermore, the same reviewer raises concerns about the novelty of the proposed position with respect to e.g. Chinese room argument. Another reviewer criticizes the paper for not discussing alternative positions.

**Questions:**

See individual reviews.

**Thoroughness:**

1

---

### Decision · Program_Chairs · 2025-09-26

Reject